# Active acetylcholine receptors prevent the atrophy of skeletal muscles and favor reinnervation

Bruno A. Cisterna [1,2,3], Aníbal A. Vargas[4], Carlos Puebla [4], Paola Fernández[2], Rosalba Escamilla[1,2], Carlos F. Lagos [5], María F. Matus[6,7], Cristian Vilos [3,8], Luis A. Cea[9], Esteban Barnafi[10], Hugo Gaete[10], Daniel F. Escobar[11], Christopher P. Cardozo[12,13] & Juan C. Sáez [1,2]

Denervation of skeletal muscles induces severe muscle atrophy, which is preceded by cellular alterations such as increased plasma membrane permeability, reduced resting membrane potential and accelerated protein catabolism. The factors that induce these changes remain unknown. Conversely, functional recovery following denervation depends on successful reinnervation. Here, we show that activation of nicotinic acetylcholine receptors (nAChRs) by quantal release of acetylcholine (ACh) from motoneurons is sufficient to prevent changes induced by denervation. Using in vitro assays, ACh and non-hydrolysable ACh analogs repressed the expression of connexin43 and connexin45 hemichannels, which promote muscle atrophy. In co-culture studies, connexin43/45 hemichannel knockout or knockdown increased innervation of muscle fibers by dorsal root ganglion neurons. Our results show that ACh released by motoneurons exerts a hitherto unknown function independent of myofiber contraction. nAChRs and connexin hemichannels are potential molecular targets for therapeutic intervention in a variety of pathological conditions with reduced synaptic neuro-muscular transmission.

[1] Departamento de Fisiología, Pontificia Universidad Católica de Chile, Santiago, Chile. [2] Centro Interdisciplinario de Neurociencias de Valparaíso, Universidad de Valparaíso, Valparaíso, Chile. [3] Centro de Investigaciones Médicas, Escuela de Medicina, Universidad de Talca, Talca, Chile. [4] Instituto de Ciencias de la Salud, Universidad de O'Higgins, Rancagua, Chile. [5] Facultad de Medicina y Ciencia, Universidad San Sebastián, Santiago, Chile. [6] Thrombosis Research Center, Medical Technology School, Department of Clinical Biochemistry and Immunohaematology, Faculty of Health Sciences, Universidad de Talca, Talca, Chile. [7] Department of Physics, Nanoscience Center (NSC), University of Jyväskylä, FI-40014 Jyväskylä, Finland. [8] Centro para el Desarrollo de la Nanociencia y Nanotecnología (CEDENNA), Universidad de Santiago de Chile, Santiago, Chile. [9] Instituto de Ciencias Biomédicas, Facultad de Ciencias de la Salud, Universidad Autónoma de Chile, Santiago, Chile. [10] Sección de Biología Molecular, Laboratorio Barnafi Krause, Santiago, Chile. [11] Sección de Biotecnología, Departamento de Salud Ambiental. Instituto de Salud Pública de Chile, Santiago, Chile. [12] National Center for the Medical Consequences of Spinal Cord Injury, James J. Peters Veterans Affairs Medical Center, Bronx, NY, USA. [13] Departments of Medicine and Rehabilitation Medicine, Icahn School of Medicine at Mount Sinai, New York, NY, USA. ✉email: bcisterna@uc.cl; jsaez@bio.puc.cl

The innervation of skeletal muscles exerts a critical influence on the maintenance of physiological tone and function[1–3]. Consequently, denervation induces severe muscle atrophy and weakness[4–6], which are preceded by a set of poorly understood cellular alterations, such as fall of resting membrane potential (RMP)[7,8], ionic imbalance[9–11], and accelerated protein catabolism[12–14].

Recently, it has been reported that denervation induces de novo expression of connexin43 (Cx43) and connexin45 (Cx45), which form connexin hemichannels (Cx HCs) in the sarcolemma of fast skeletal myofibers[15,16], cause increased sarcolemmal permeability[15,16], and favor an ionic imbalance during muscle atrophy[16] that ultimately lead to atrophy. The importance of Cxs during early stages of myogenesis is rather well established[17]. However, there is also considerable evidence that innervation and/or neuromuscular activity results in their down-regulation during adulthood through an unknown molecular mechanism[18–20]. The Cx HCs are none selective plasma membrane channels that allow the passage of ions driven by their electrochemical gradients[21], and small molecules, such as ATP[22,23], NAD$^+$[24], and glutamate[25].

Pioneering investigations in denervated skeletal muscles showed that there is a direct relationship between the length of the nerve stump and time course of failure of the stump to transmit impulses to the muscle[26–28]. Specifically, the ability to transmit impulses is prolonged by about 45 min for each additional centimeter in the nerve stump[29]. These findings suggest that there is transport and release of protective factor(s) from the nerve stump, which are ultimately depleted as axonal reserves are consumed over time. This hypothesis is reinforced by clinical observation in an accidental overdose of vincristine, an axonal transport blocker, where muscle weakness and atrophy was observed[30]. In this sense, neuron-derived factors may be responsible for maintaining the normal innervated skeletal muscle phenotype; such factors may include acetylcholine (ACh), adenosine triphosphate (ATP), or neurotrophins such as brain-derived neurotrophic factor (BDNF) or nerve growth factor (NGF). The primary purpose of this study was to identify the neuron-derived factor(s) responsible for suppressing Cx HCs expression in innervated myofibers.

In order to evaluate the effect of the different factors released by the nerve, we use cultured skeletal myofibers as an in vitro denervated model, which we demonstrate herein recapitulated the genetic and biochemical alterations found in denervated muscles in vivo. These alterations include increased expression of atrophy genes called "atrogenes" atrogin-1 (Fbxo32) and MuRF1 (Trim63)[31,32], and increased expression of the autophagy gene Bnip3[33], without alterations of pro-apoptotic genes and proteins[34,35].

In this work, we identify ACh as the nerve-derived molecule responsible for preventing the expression of Cx43 and Cx45 HCs in the sarcolemma and thus, prevent the increase of sarcolemmal permeability, the fall of RMP, and the increase of Ca$^{2+}$ and Na$^+$ signals in cultured skeletal myofibers. Consistently with these findings, the in vivo increase in the half-life of ACh decreased the atrophy in denervated mice. Likewise, the protective effect of ACh is carried out by the nicotinic acetylcholine receptors (nAChRs) through a post-transcriptional mechanism involved in controlling Cxs mRNA translation.

Collectively, our data support the conclusion that ACh released by motoneurons, through nAChR, prevents the occurrence of connexin-mediated atrophy. Likewise, we identified that the activity of Cx43 and Cx45 HCs of sarcolemma is detrimental to reinnervation and nearby axons.

## Results

### ACh prevents the denervation-induced cellular alterations.
Innervation by motor neurons is key to maintain the normal

phenotype of skeletal myofibers[4–6]. However, how the motor neuron exerts its protective effect on the myofiber remains unknown. Since the evaluation of the involvement of different neuronal derived factors in vivo could cause undesired effects beyond the neuromuscular junction (NMJ), we used primary cultures of dispersed myofibers to evaluate whether neuron-derived factors prevent the phenotypic changes that characterize denervated skeletal myofibers. We first validated this in vitro system as a model of muscle atrophy by measuring the relative levels of mRNAs of molecular markers of atrophy, autophagy, and apoptosis. To accomplish this, we used mice flexor digitorum brevis (FDB) myofibers cultured for 72 h to determine if they undergo molecular changes compatible with the response to atrophy observed in vivo. We evaluated the expression of atrogenes (Fbox32 and Trim63) and autophagy genes (Bnip3), which are altered in vivo in denervated FDB muscles[31,33]. In addition, we evaluated the expression of the pro-apoptosis gene (Bak1) and compared it to the anti-apoptotic gene (Bcl2)[34,35]. All genes were normalized as fold expression relative to β-actin.

In freshly isolated myofibers (in vitro, 0 h), the mRNA levels of FBox32, Trim63, and Bnip3 were comparable to those of in vivo innervated (Inne) FDB muscles (Fig. 1a). Likewise, the levels of these mRNAs showed a marked increase in myofibers cultured for 72 h as well as in FDB muscles after 7 days of denervation (Den) (Fig. 1a). These findings are consistent with pioneering research on gene expression of denervated skeletal muscle[31,33]. In contrast, the levels of mRNAs of FBox32, Trim63, and Bnip3 in myofibers treated with carbachol (Cbc), a nicotinic acetylcholine receptor (nAChR) agonist[36], applied every day up to 72 h were comparable to the levels of each mRNA detected in innervated FDB muscles or in freshly isolated myofibers (Fig. 1a). In addition, we observed increased atrogin-1 and μ-calpain immunoreactivity after 48 and 72 h of culture under control conditions (Fig. 1b). These changes were not observed in myofibers treated with Cbc (Fig. 1b).

With regard to the levels of pro-apoptotic (Bak1) and anti-apoptotic (Bcl2) markers, in innervated (Inne) muscles, we observed a decrease in the levels of Bak1 mRNA and an increase in Bcl2 mRNA (Fig. 1c), which resulted in a decreased Bak1/Bcl2 ratio (Fig. 1d), whereas in denervated (Den) muscles, an increased mRNA level of both, Bak1 and Bcl2, with a greater increase in Bak1 (Fig. 1c), yielded an increased Bak1/Bcl2 ratio (Fig. 1d), as reported previously for denervated skeletal muscles[34,35]. In cultured myofibers, at 0 h of culture we found decreased mRNA levels of both, Bak1 and Bcl2, with a greater decrease in Bak1 (Fig. 1c), giving a decreased Bak1/Bcl2 ratio (Fig. 1d). At 72 h of culture, we found decreased levels of mRNA of Bak1 and increased of Bcl2 (Fig. 1c), giving a decreased Bak1/Bcl2 ratio (Fig. 1d). Myofibers treated for 72 h with Cbc showed decreased Bak1 and increased Bcl2 mRNA levels (Fig. 1c), with a decreased Bak1/Bcl2 ratio (Fig. 1d). On the other hand, in myofibers treated for 72 h with Paclitaxel used as positive control for apoptosis[37], the levels of Bak1 mRNAs increased and Bcl2 decreased (Fig. 1c), resulting in an increased Bak1/Bcl2 ratio (Fig. 1d). In addition, we found similar anexin V and caspase 3 immunoreactivities at 0 h and after 72 h under control conditions, as well as after 72 h treatment with Cbc, whereas myofibers treated for 72 h with Paclitaxel showed an increase in caspase 3 and anexin V immunoreactivity (Fig. 1e). Thus, the above findings revealed that cultured myofibers undergo atrophy without apoptosis similar to what occurs in vivo denervated muscles, validating the culture of primary myofibers as a model of muscle denervation.

We next evaluated effects of nerve-derived molecules or their synthetic analogs on increased sarcolemmal permeability to ethidium (Etd$^+$) bromide as a measure of de novo sarcolemmal

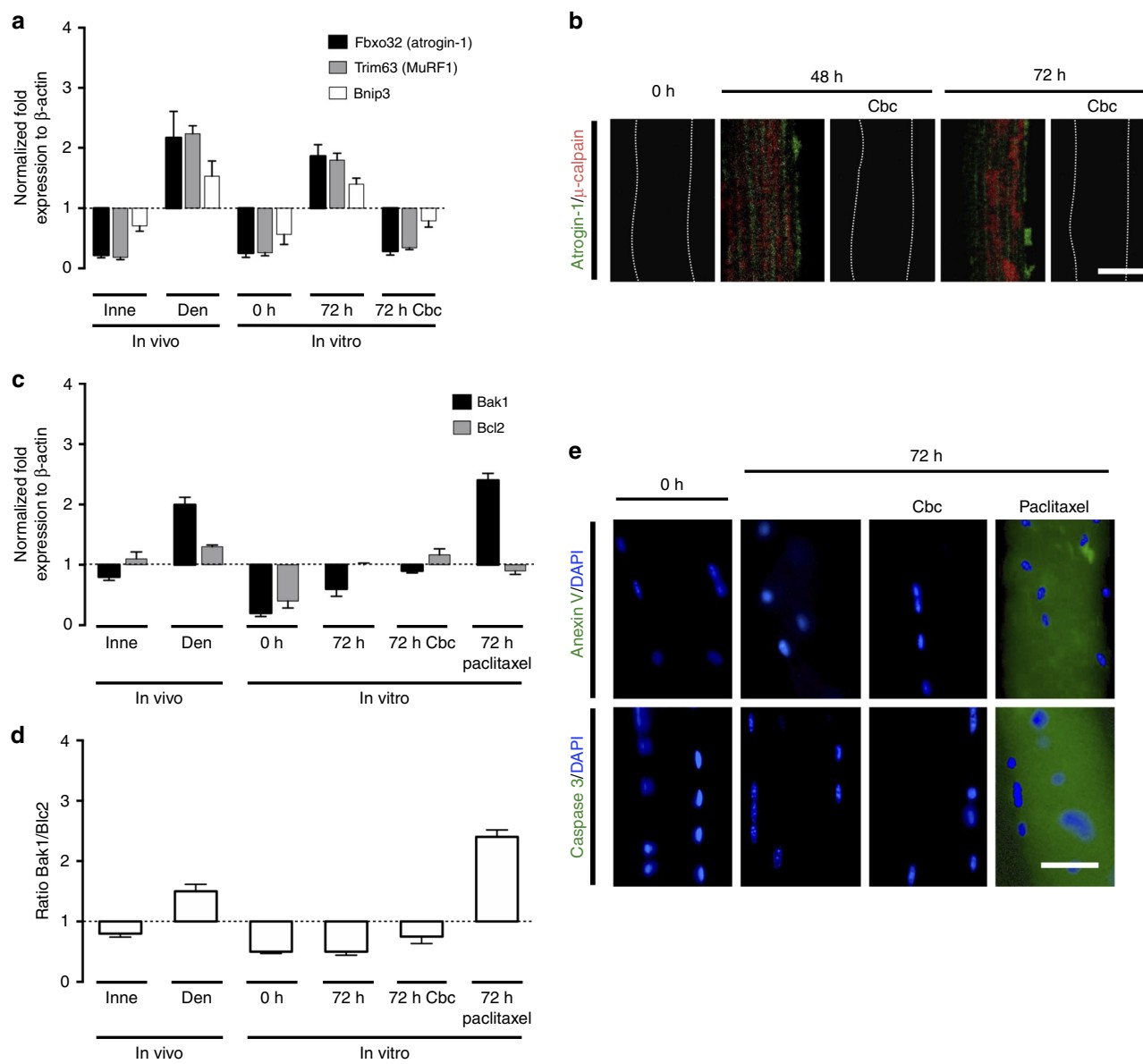

**Fig. 1 Cultured fibers express genes of atrophy and autophagy, but not of apoptosis.** Unilateral sciatic nerve transections were performed in $Cx43^{fl/fl}Cx45^{fl/fl}$ mice (control mice); at day 7 post denervation the innervated (Inne) and denervated (Den) flexor digitorum brevis muscles were collected to determine the relative levels of specific mRNAs by qPCR (In vivo). In parallel experiments, myofibers from control mice were cultured for 0, 48, and 72 h under control conditions or treated with 200 nM carbachol (Cbc) or 100 nM Paclitaxel to determine the relative levels of specific mRNAs by qPCR and proteins by immunofluorescence (In vitro). **a** mRNA relative levels of Fbxo32 (atrogin-1), Trim63 (MuRF1), and Bnip3 normalized to actin β expression. **b** Relative levels of atrogin-1 (Green) and μ-calpain (Red) evaluated by immunofluorescence using confocal microscopy. Calibration bar: 50 μm. **c** mRNA relative levels of Bak1 (pro-apoptotic gene) and Bcl2 (anti-apoptotic gene) normalized to actin β expression. **d** Ratio of Bak1/Bcl2. **e** Relative levels of anexin V or caspase 3 (Green) evaluated by immunofluorescence (DAPI to stain nuclei in blue). Calibration bar: 50 μm. Bar error is the mean ± SEM. $N = 4$ independent experiments.

Cx HCs expression. Specifically, we isolated and cultured myofibers from FDB muscles and treated them as follows: to evaluate the effect of ATP, which in vivo is co-released with ACh[38], we applied ATP every 6 h, or adenosine 5′-O-(3-thio) triphosphate (ATP-γ-S), a non-hydrolyzable ATP analog, every 24 h. To evaluate the effect of nAChR agonizts we applied Cbc, or methyl-carbamylcholine (Cmc), two non-hydrolyzable synthetic ACh analogs, every 24 h or the natural ligand ACh every 6 h. Since neurons release neurotrophins that affect diverse muscle characteristics[39,40] and in primary culture the source of these factors is very limited, we decided to evaluate the effect of exogenous neurotrophins. To this end, we used BDNF or NGF applied alone or simultaneously every 24 h. Control cultured

myofibers exhibited increased sarcolemmal permeability at 48 h of culture (Fig. 2a, b). However, the increase in sarcolemmal permeability was prevented entirely by Cbc (Fig. 2a, b), ACh or Cmc (Supplementary Fig. 1), whereas treatment with ATP, ATP-γ-S, NGF, BDNF, or NGF + BDNF had no effect (Fig. 2a, b and Supplementary Fig. 1).

Because Cbc prevented the increase in sarcolemmal permeability and the increase of expression of atrogenes, we evaluated it's effect on other alterations. After 48 and 72 h of culture under control conditions or after treatment with ATP or NFG + BDNF, myofibers showed reduction in RMP from −75 to −55 mV (about 30%) (Fig. 2c), similar to that found in vivo from 5 days post-denervation (Supplementary Fig. 2), and concordant with

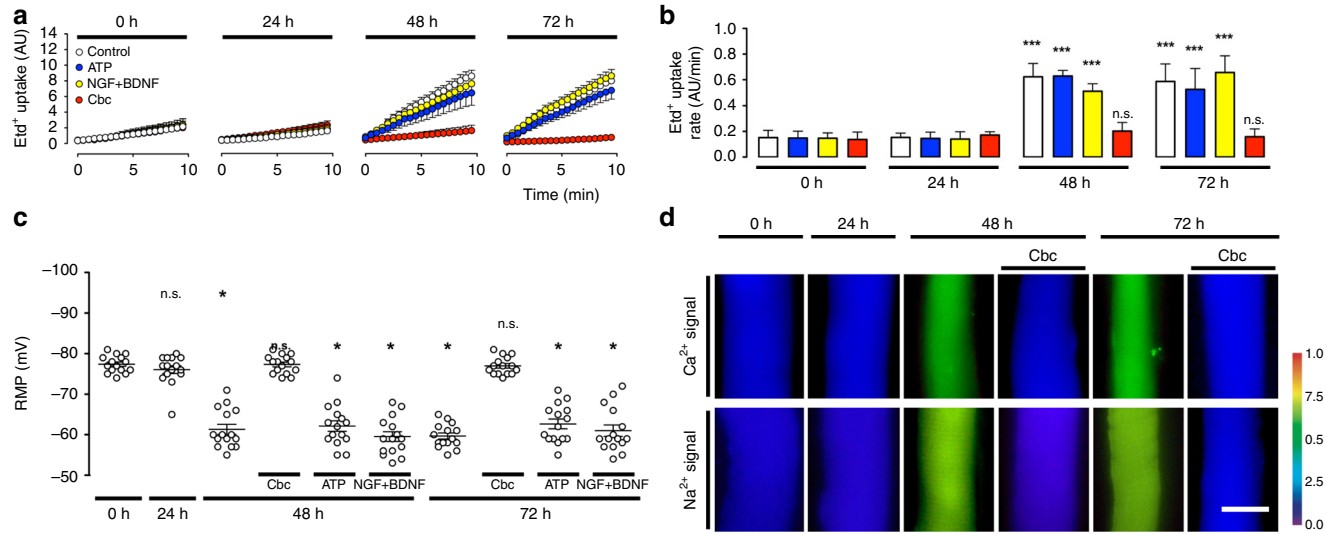

**Fig. 2 Acetylcholine analog prevents cellular alterations in cultured myofibers. Primary cultures of myofibers from flexor digitorum brevis (FDB) muscles were used. a** The permeability of the sarcolemma was measured in time-lapse experiments of ethidium (Etd$^+$) uptake performed after 0, 24, 48, and 72 h of culture. Myofibers were cultured under control conditions (white), or treated with 500 μM ATP (blue) or 50 ng/mL + NGF/50 ng/mL BDNF (NGF + BDNF; yellow), or 200 nM Cbc (red). **b** Etd$^+$ uptake rate of myofibers. $N = 4$ independent experiments; six myofibers were recorded in each experiment, each value is the mean ± SEM. ***$p < 0.001$ compared with myofibers at 0 h of culture; n.s. non-significant difference, by ANOVA with Bonferroni post hoc test. **c** Resting membrane potential (RMP) was evaluated at 0, 24, 48, and 72 h of culture. Myofibers were cultured under control conditions, or treated with ATP, NGF/BDNF, or Cbc. $N = 5$ independent experiments with at least twenty myofibers recorded in each independent experiment. *$p < 0.05$, compared with myofibers at 0 h of culture by ANOVA with Bonferroni post hoc test. **d** Upper panel, intracellular Ca$^{2+}$ signal (340/380), lower panel, intracellular Na$^+$ signal was recorded at 0 and 48 h of culture using FURA-2 or SBFI, respectively. The colored scale to the right of the panels depicts the color shifts from blue to green as the dye is bound to Ca$^{2+}$ or Na$^+$. Parallel cultures were treated at time 0 with 200 nM carbachol (Cbc) and 48 h later the Ca$^{2+}$ and Na$^+$ signal was evaluated· Parallel cultures were treated at time 0 with 200 nM carbachol (Cbc) and 48 h later the Ca$^{2+}$ and Na$^+$ signal was evaluated· Scale bar: 50 μm.

previously reported at different times in myofibers from different species[7,8,41]. In addition, we observed increased intracellular Ca$^{2+}$ and Na$^+$ signals after 48 and 72 h of culture under control conditions. These changes were not observed in myofibers treated with Cbc (Fig. 3d). Of note, treatment with Cbc did not reverse cellular alterations (RMP and sarcolemmal permeability) once these were already present (Supplementary Fig. 3). In conclusion, the lack of nervous supply in cultured myofibers induces alterations equivalent to those observed in denervated muscles in vivo. Likewise ACh analogs prevent these alterations in vitro.

Next, we administered daily (for 6 days) subcutaneous pyridostigmine (Pyri) bromide, an acetylcholinesterase inhibitor that increases the half-life of ACh[42], to mice after unilateral sciatic nerve transection (Fig. 3a). At day 7 post denervation, there was no significant decrease in denervated muscle fiber size compared to innervated muscle. By contrast, denervated muscle showed significant muscular atrophy (about 50%) at 7 days post denervation in controls not treated with Pyri (Fig. 3b, c). Therefore, administering ACh prevents cellular alterations in vitro, and increasing its half-life prevents decreasing myofiber size in vivo, all of which demonstrates that ACh exerts a protective effect on skeletal myofibers (Fig. 3d).

It is worth mentioning that according to molecular docking analysis, both synthetic ACh analogs used bind to the α subunit of nAChR, Cbc binds between α$_γ$ and γ subunits, the same site as ACh, while Cmc binds between the α$_δ$ and γ subunits (Supplementary Fig. 4). Therefore, the effect observed in vitro with ACh, Cbc and Cmc may be exerted by ACh in vivo. Thus, the continuous availability of ACh under physiological conditions preserves the normal phenotype of skeletal myofibers through an as yet unknown molecular mechanism (Fig. 3d).

**Involvement of nAChR in the protective effect of ACh**. We then evaluated whether the protective effect of ACh analogs was achieved through nAChR. In myofibers treated with pancuronium (Pcu), a competitive blocker of the nAChR[43], the increase in membrane permeability (Fig. 4a, b), decrease in RMP (Fig. 4c), and increase in Ca$^{2+}$ and Na$^+$ signals (Fig. 4d) were already evident at 24 h of culture. Furthermore, Pcu blocked excitatory miniature end-plate potentials (MEPPs) in the NMJ (Supplementary Fig. 5). Therefore, by accelerating the alterations of the myofibers by inhibiting the nAChR, we demonstrate a hitherto unknown, protective effect of ACh through the nAChR (Fig. 4e). These results explain pioneering research showing a direct relationship between the length of the axonal stump and the time course of failure to transmit impulses to muscle[26–28,44]. We propose that the timing observed depends on the number of synaptic vesicles containing ACh in the nerve stump, which are released and consumed over time.

During adulthood, the activity of nAChR only has been related to the conversion of neuronal chemical signals into a sarcolemmal depolarization at NMJs to produce a mechanical response in muscle. In order to identify the signaling pathway associated with the nAChR activity involved in the protective effect of ACh on denervated myofibers maintained in culture, we evaluated the possible involvement of protein kinases that have been shown to be downstream of AChR and reported altered after denervation: PKC and PKA[45], P38 MAPK[46], and Rho[47]. We tested effects of the following blockers of these kinases on sarcolemmal permeability of cultured myofibers: Calphostin C (Calpho C; a protein kinase C inhibitor), Bisindolylmaleimide (Bisindolyl; a protein kinase C inhibitor), H-89 dihydrochloride hydrate (H-89; a protein kinase A inhibitor), SB203580 (a p38 MAPK inhibitor), and Fasudil (a Rho-kinase inhibitor). In addition, we evaluated

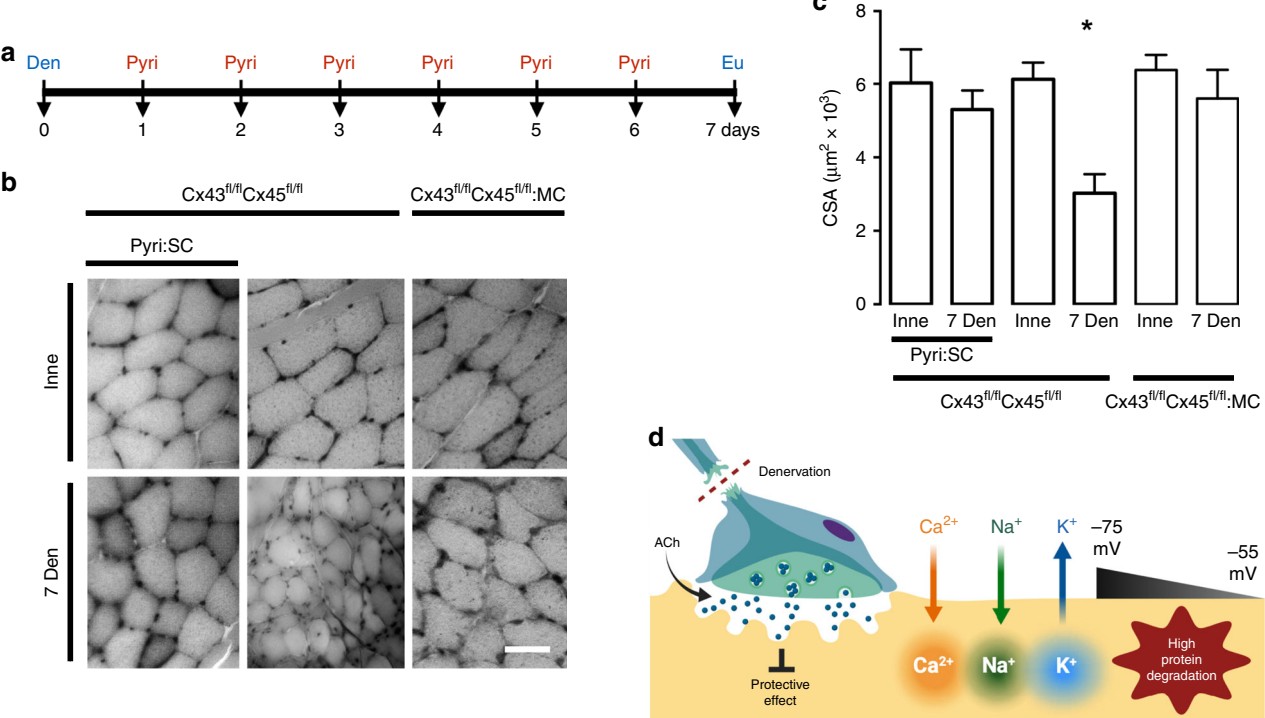

**Fig. 3 Acetylcholine analog prevents cellular alterations in vitro, and increasing its half-life prevents the decrease of myofiber size in vivo.** Unilateral denervation of the sciatic nerve in Cx43[fl/fl]Cx45[fl/fl]:Myo-Cre (Cx43[fl/fl]Cx45[fl/fl]:MC) mice were performed. **a** Experimental design of in vivo acetylcholinesterase blockade. Den: denervation, Pyri: pyridostigmine, Eu: euthanasia. **b** Hematoxylin:eosin-stained cross-section of the FDB at day 7 post denervation. Inne: Innervated. Den: Denervated. **c** Cross-sectional area (CSA). $N = 5$; each value is the mean ± SEM. *$p < 0.05$ for Den compared with Inne muscles by Student's $t$ test. **d** Proposed model. Denervation (Den) eliminates the protective effect of ACh and leads to a reduction in acetylcholine (ACh) release resulting in an increase in $Ca^{2+}$ and $Na^+$ influx via de novo expressed non-selective membrane channels, increasing cytoplasmic concentrations of these ions. Consequently, a reduction in the RMP from −75 to −55 mV occurs due in part by $K^+$ efflux and $Ca^{2+}$ and $Na^+$ influx; the increase in intracellular $Ca^{2+}$ signal promotes protein degradation.

the effect each blocker combined with Cbc to determine which is upstream of the other.

Only in the presence of Calpho C and Bisindolyl was increase in sarcolemmal permeability hastened from 48 to 24 h (Supplementary Fig. 6a, b). The co-incubation of these molecules with Cbc (Cbc + Calpho C and Cbc + Bisindolyl) prevented the inhibitory effect of Cbc on sarcolemmal permeability increase at 48 h of culture, unlike the other molecules co-incubated with Cbc (Supplementary Fig. 6c, d). Thus, we suggest that activation of PKC is downstream of nAChR, and may participate in the signal transduction of the protective effect of ACh.

The connection between AChR and PKC has been well documented in the literature. AChRs are recognized to have phosphorylation sites for at least three protein kinases, such as PKC, PKA, and PTK[48,49]. Although PTK phosphorylation of nAChR is suggested to accelerate the rate of the current decay time, the effect of PKC and PKA phosphorylation of the receptors also increases the rate of desensitization[50]. Likewise, the balance between PKA and PKC activities may be critical for the maintenance of postsynaptic receptor density[45].

In our studies, we found that normal fast skeletal muscle express Cx mRNAs in myofibers and 7 days of denervation does not induce significant changes in Cx43 and Cx45 mRNA levels (Supplementary Fig. 6e, f). Likewise, freshly isolated myofibers already express Cx43 and Cx45 mRNA, and 48 h of culture induces an increase in Cx45 mRNA only, while Cbc treatment significantly decreases both mRNA connexin relative levels (Supplementary Fig. 6g, h). By contrast, the immunoreactivity of Cx43 and Cx45 is very low in normal innervated muscles

(western blot analysis, Supplementary Fig. 6i–l) and most likely reflects a contribution from in cell types different form myofibers as vascular cells[51], since in the later the connexin immunofluorescence detection was negative (Supplementary Fig. 6m), whereas denervation induces significant increases (Supplementary Fig. 6i–m). Therefore, a post-transcriptional mechanism is involved in controlling Cxs mRNA translation. Moreover, we observed that blockade of lysosome- or ubiquitin proteasome-dependent protein degradation pathway (chloroquine and G5 ubiquitin isopeptidase inhibitor, respectively) does not affect the increase in either Cx43/Cx45 immunoreactivities or sarcolemma permeability in cultured skeletal myofibers. However, blockade of protein synthesis with 100 μg/ml cycloheximide for 48 h prevents both alterations (Supplementary Fig. 7), suggesting that activation of nicotinic AChRs lead to inhibition of protein synthesis of Cxs.

It is worth mentioning that unilateral immobilization of skeletal muscles, a condition in which the quantal ACh supply is normal, we observed atrophy (Supplementary Fig. 8a, b) and a clear increase in MyoD relative levels (Supplementary Fig. 8c), an indicator of immobilization-induced atrophy[52], but the relative levels of Cx43 and Cx45 in immobilized muscles remained low and were comparable to those of control muscles (contralateral) (Supplementary Fig. 8d, e). Therefore, the posttranscriptional mechanism involved in controlling Cxs mRNA translation is related to ACh and its signaling.

**ACh analogs repress the expression of Cx HCs.** Since our data show that the absence of ACh initiates a set of cellular alterations

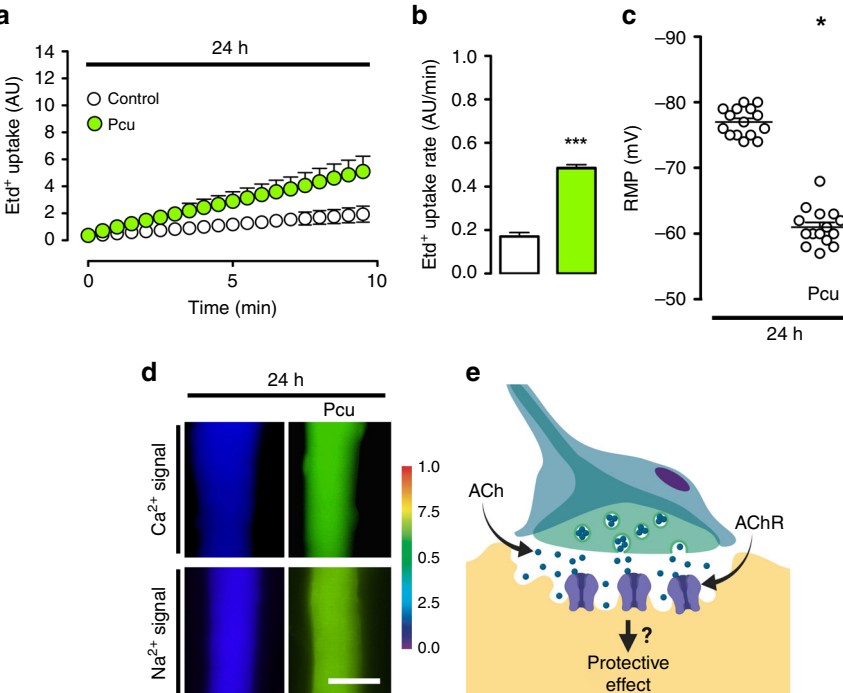

**Fig. 4 The activity of nicotinic acetylcholine receptor (nAChR) prevents atrophy of skeletal myofibers.** Primary cultures of myofibers from flexor digitorum brevis (FDB) muscle were used. **a** Membrane permeability was measured in time-lapse experiments evaluating ethidium (Etd$^+$) uptake performed after 24 h of culture. Myofibers were cultured under control conditions (white), or treated with 15 μM pancuronium (Pcu; green). **b** $N = 4$; six myofibers recorded in each independent experiment, each value is the mean ± SEM. ***$p < 0.001$, for effect of Pcu compared with Control by Student's $t$ test. **c** Resting membrane potential (RMP) was evaluated at 24 h of culture. Myofibers were cultured under control conditions or after treatment with 15 μM Pcu. $N = 5$; at least 20 myofibers recorded in each independent experiment. *$p < 0.05$, for effect of Pcu compared with Control by Student's $t$ test. **d** Upper images, intracellular Ca$^{2+}$ signals (340/380) and, lower images, Na$^+$ signals were recorded at 24 h of culture with FURA-2 and SBFI, respectively. The colored scale to the right of the panels depicts the color shifts from blue to green as the dye is bound to Ca$^{2+}$ or Na$^+$. Parallel cultures were treated with 15 μM Pcu and 24 later the Ca$^{2+}$ and Na$^+$ signals were evaluated. Scale bar: 50 μm. **e** Proposed model.

characteristic of denervation, we evaluated the molecular mechanism responsible for these alterations. Recently, the de novo expression of Cx43 HCs and Cx45 HCs in the sarcolemma of denervated myofibers in vivo was demonstrated to explain increased sarcolemma permeability, intracellular Ca$^{2+}$ and Na$^+$ signals and protein catabolism observed after denervation[15,16]. Of note, these membrane permeability changes are not present in muscles during immobilization-induced atrophy, a condition in which the quantal ACh supply is normal (Supplementary Fig. 8). Therefore, it was necessary to determine if the protective effect of ACh is due to the prevention of Cxs expression. To this end, we evaluated the Etd$^+$ uptake rate in cultured myofibers from Cx43$^{fl/fl}$Cx45$^{fl/fl}$ mice (control mice) and Cx43$^{fl/fl}$Cx45$^{fl/fl}$:MC mice (mice lacking Cx43 and Cx45 in skeletal muscles).

At 48 h of culture under control conditions or treatment with NGF + BDNF or ATP, the myofibers exhibited increased Etd$^+$ uptake rate; acute treatment with the Cx HC blockers La$^{3+}$, carbenoxolone (Cbx), flufenamic acid (FFA) or D4 reduced Etd$^+$ uptake (Supplementary Figs. 9 and 10). Treatment with Cbc completely prevented the increase in sarcolemmal permeability, as did conditional knockout of Cx43/45 in Cx43$^{fl/fl}$Cx45$^{fl/fl}$:MC mice (Fig. 5a, b). Similarly, immunoreactivity to Cx43 and Cx45 increased at 48 h of culture, and treatment with Cbc prevented that increase (Fig. 5c). Also, we evaluated the possible participation of other channel-forming proteins using KO mice lacking either Cx43, Cx45, Panx1 or P2X$_7$, and found that the absence of these proteins individually did not significantly prevent denervation-induced muscle atrophy (Supplementary Fig. 11). Accordingly, the increase in sarcolemma permeability is due to

the simultaneous expression of Cx43 and Cx45, which is prevented by ACh.

We also evaluated RMP in cultured myofibers in vitro (Supplementary Figs. 2, 12 and 13). In denervated FDB muscles from Cx43$^{fl/fl}$Cx45$^{fl/fl}$ mice, RMP decreased from 5 days post-denervation; this reduction was reversed by 90 min treatment with La$^{3+}$, Cbx or D4, a computationally designed selective Cx HCs blocker, and was not observed in myofibers of Cx43$^{fl/fl}$Cx45$^{fl/fl}$:MC mice (Supplementary Fig. 2). Interestingly, the recovery of RMP observed at 90 min with Cx HC blockers was reduced by Ouabain (OB), a Na$^+$/K$^+$ ATPase pump blocker, indicating that once the Cx HCs are blocked, the recovery of the membrane potential is carried out by the Na$^+$/K$^+$ ATPase pump (Supplementary Figs. 12 and 13). Accordingly, the decrease in RMP is due to the expression of Cx43 and Cx45 and ACh prevents their expression. To validate the use of transgenic mice and avoid any compensatory response during muscle development, we silenced the same proteins in cultured myofibers from control mice, using morpholinos against Cx43 and Cx45 (Cx43 + Cx45 Mor). At 48 h of treatment with Cx43 + Cx45 Mor, the increase in both sarcolemma permeability and immunoreactivity of Cx43 and Cx45 were prevented (Fig. 5d–f).

**Cx HCs are harmful to innervation and nearby axon growth.** The peripheral nervous system has the intrinsic capacity to regenerate. However, the success of reinnervation of skeletal myofibers is widely variable for reasons that are not fully understood[53–55]. To know if sarcolemmal Cx HCs can alter

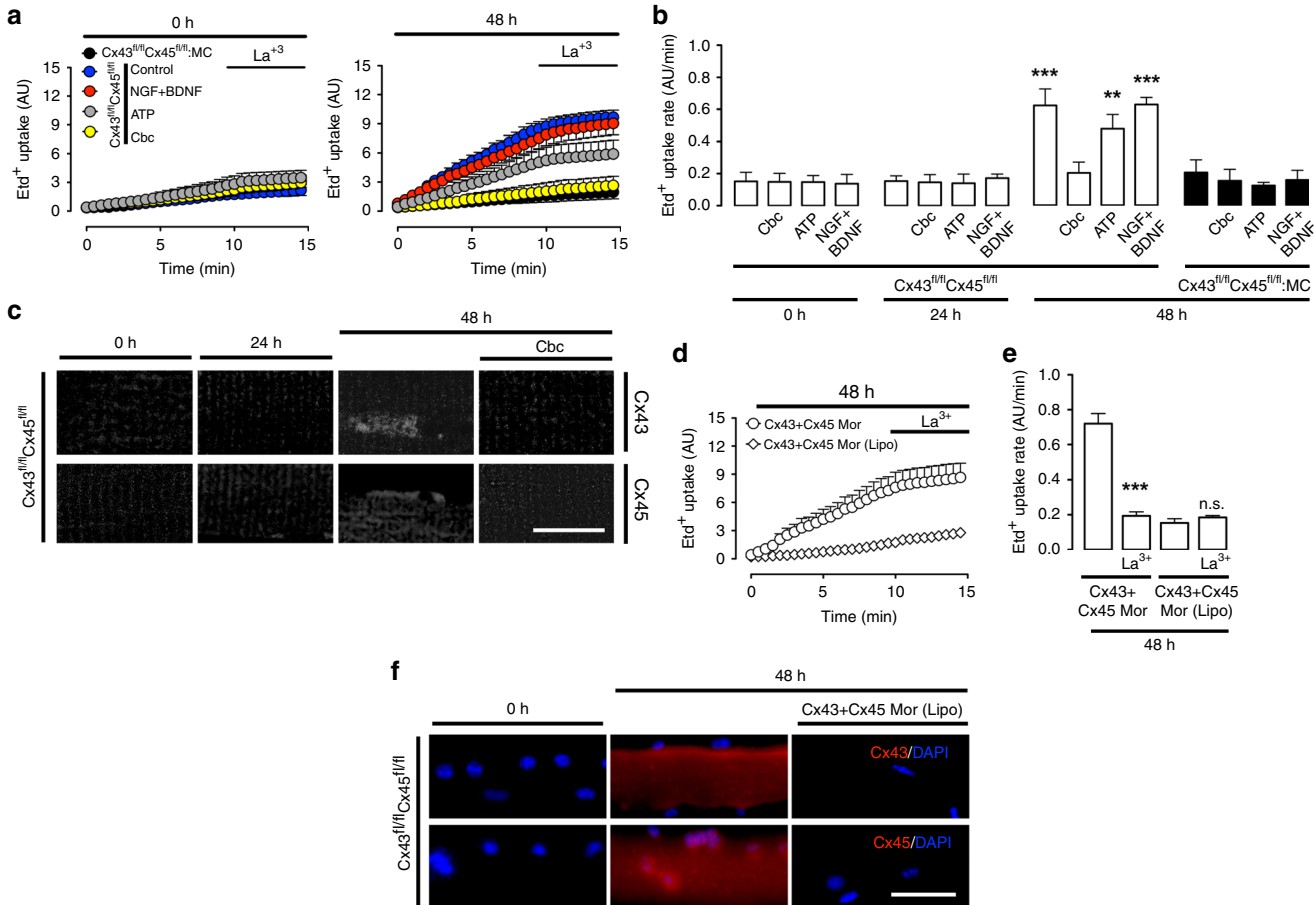

**Fig. 5 Acetylcholine analogs repress the expression of connexin hemichannels, which leads to atrophy in denervated skeletal muscles.** Primary cultures of myofibers from flexor digitorum brevis muscles of $Cx43^{fl/fl}Cx45^{fl/fl}$ and $Cx43^{fl/fl}Cx45^{fl/fl}$:Myo-Cre ($Cx43^{fl/fl}Cx45^{fl/fl}$:MC) mice were used. **a** Membrane permeability was evaluated in ethidium ($Etd^+$) uptake experiments at 0 h (left) or 48 h (right) of culture. The first 10 min are baseline, after which myofibers were treated with 200 μM $La^{3+}$. The sarcolemmal permeability of myofibers from $Cx43^{fl/fl}Cx45^{fl/fl}$ was measured in culture under control conditions (blue circles), or in the presence of 50 ng/mL NGF + 50 ng/mL BDNF (NGF/BDNF; red circles), 500 μM ATP (gray circles), or 200 nM Cbc (yellow circles). Myofibers of $Cx43^{fl/fl}Cx45^{fl/fl}$:MC mice were cultured under control conditions (black circles). **b** $Etd^+$ uptake rate of myofibers. $N = 4$; five myofibers recorded in each independent experiment, each value is the mean ± SEM. ***$p < 0.001$, and **$p < 0.005$ compared to myofibers at 0 h of culture by ANOVA with Bonferroni post hoc test. Data obtained from $Cx43^{fl/fl}Cx45^{fl/fl}$ mice are represented by white bars, and from $Cx43^{fl/fl}Cx45^{fl/fl}$:MC mice by black bars. **c** Relative levels of Cx43 and Cx45 evaluated by immunofluorescence using confocal microscopy at 0, 24, and 48 h of culture. Myofibers cultured for 48 h were also treated at time 0 with 200 nM Cbc. Calibration bar: 100 μm. **d**, $Etd^+$ uptake at 48 h of culture. Myofibers were treated with Cx43 + Cx45 morpholinos (Cx43 + Cx45 Mor) or Cx43 + Cx45 morpholinos with Lipofectamine® 2000 [(Cx43 + Cx45 Mor (Lipo)]. The first 10 min are baseline then myofibers were treated with 200 μM $La^{3+}$. **e** $Etd^+$ uptake rate of myofibers. $N = 4$; five myofibers recorded in each independent experiment, each value is the mean ± SEM. ***$p < 0.001$, for effect of $La^{3+}$ on Cx43 + Cx45 Mor compared with Cx43 + Cx45 Mor. n.s. non-significant difference, for $La^{3+}$ on Cx43 + Cx45 Mor (Lipo) compared Cx43 + Cx45 Mor (Lipo), by ANOVA with Bonferroni post hoc test. **f** Relative levels of Cx43 and Cx45 (red) evaluated by immunofluorescence in myofibers treated with Cx43 + Cx45 Mor (Lipo) for 48 h. DAPI (blue). Calibration bar: 50 μm.

reinnervation capacity, we evaluated innervation during co-culture of dorsal root ganglion (DRG) explants and skeletal myofibers from $Cx43^{fl/fl}Cx45^{fl/fl}$ and $Cx43^{fl/fl}Cx45^{fl/fl}$:MC mice.

The DRGs were seeded on the central compartment of Campenot Chambers (0 days DRGs) after 5 days of culture (5 days DRGs), we added myofibers over distal axons on lateral compartments (0 days of co-culture) (Fig. 6a and Supplementary Fig. 14). The functional expression of Cx HCs in myofibers at 5 days of co-culture (Supplementary Fig. 15) decreased the contacts between neurites and muscle (hereafter referred to as innervation), evaluated as the percentage of myofibers contacting DRG neurons (Fig. 6b, c). Also, Cx HCs reduced numbers of dendrites observed near myofibers (Fig. 7a, b). These effects were not detected or were significantly reduced in cultured myofibers in the presence of morpholinos to Cx43 and Cx45 (Cx43 + Cx45

Mor), or myofibers from $Cx43^{fl/fl}Cx45^{fl/fl}$:MC mice, strongly suggesting that sarcolemmal Cx43 and 45 HCs in myofibers are harmful to nearby neurons, and that their inhibition could improve the reinnervation of skeletal muscles. Therefore, these results demonstrate a detrimental effect, hitherto unknown, of the functional expression of Cx43 and Cx45 HCs on myofiber-related axons. Consequently, Cx HCs could be a therapeutic target to improve the reinnervation of skeletal muscles.

Collectively, our data support our hypothesis that a neuron-derived factor is key to maintaining the normal phenotype of adult skeletal myofibers. Specifically, ACh through the activity of nAChR inhibits the Cx HCs expression in the myofibers, resulting in multiple deleterious cellular alterations, and is detrimental to innervation and nearby axons. This provides a feasible approach to identify the factors released by denervated

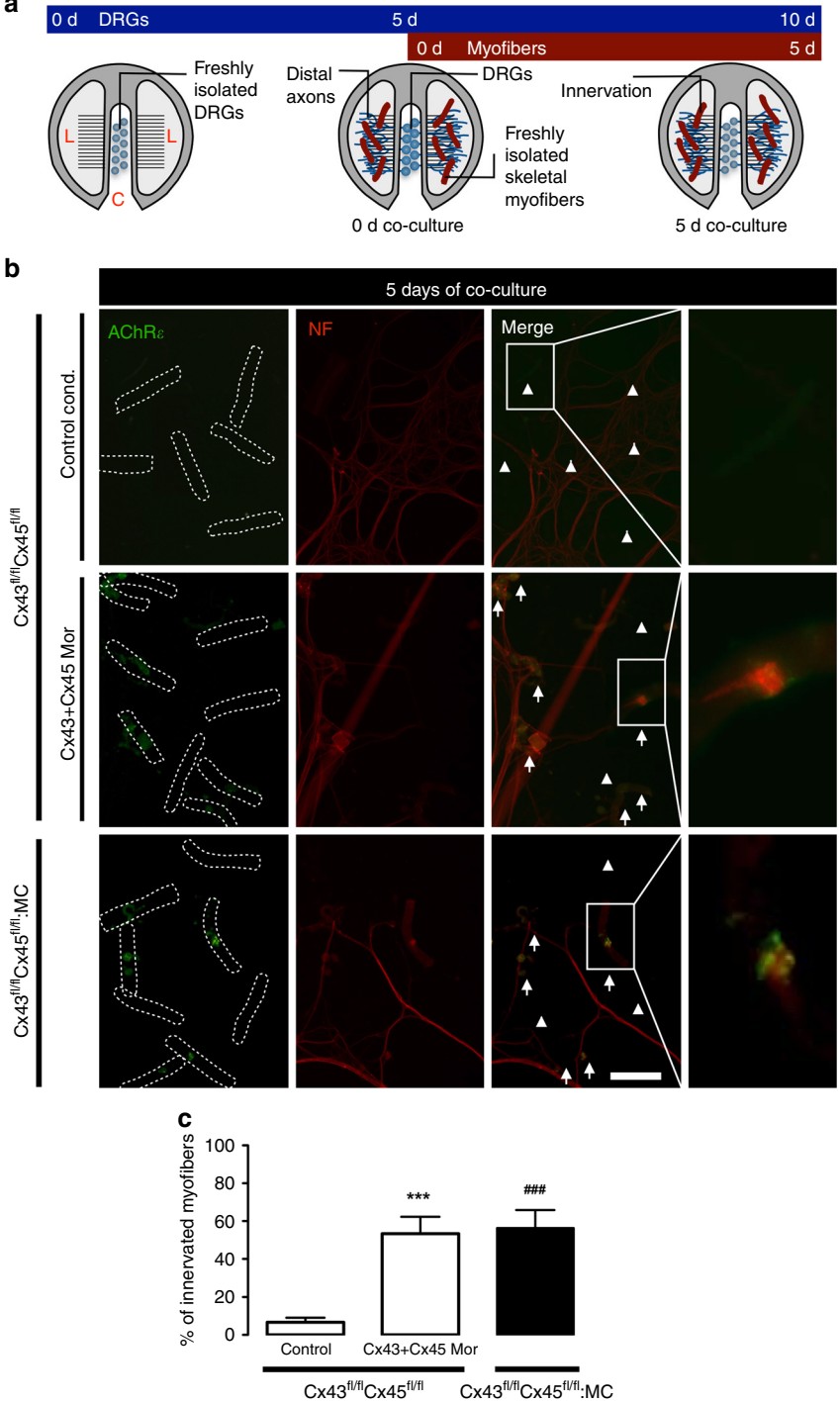

**Fig. 6 The absence of Cx43 and Cx45 in cultured skeletal myofibers increases the percentage of innervation in co-culture. DRG explants were co-cultured with skeletal myofibers. a** DRG explants were cultured in Campenot chambers in the central compartment (C), at 5 days of DRGs culture; skeletal myofibers are added in the lateral compartments (L) together with the distal axons (0 days of co-culture), up to 10 days of DRGs culture (5 days of co-culture). **b** Immunofluorescence of myofibers from Cx43fl/flCx45fl/fl mice under control conditions (Control cond.; top row), or treated with morpholinos to Cx43 and Cx45 (Cx43 + Cx45 Mor; middle row), and Cx43fl/flCx45fl/fl:MC mice (down row) at 5 days of co-culture of DRGs and myofibers, using antibodies against neurofilament 200 (NF; red; left column), and nicotinic acetylcholine receptor epsilon subunit (AChRε; green; middle left column). Merged images to identify innervated myofibers (middle right column). Arrows: innervated myofibers, arrowheads: non-innervated myofibers. Scale bar: 100 μm. Dotted rectangle is the magnification of innervated myofibers (right column). **c** Percentage of innervated myofibers. $N = 5$; at least 35 myofibers recorded in each independent experiment, each value is the mean ± SEM. The results obtained from Cx43fl/flCx45fl/fl mice are represented by white bars and from Cx43fl/flCx45fl/fl:MC mice by black bars. ***$p < 0.001$, for Cx43 + Cx45 Mor compared with Control cond. ###$p < 0.001$, for 43fl/flCx45fl/fl:MC compared with Control cond. by two-way ANOVA with Bonferroni post hoc test.

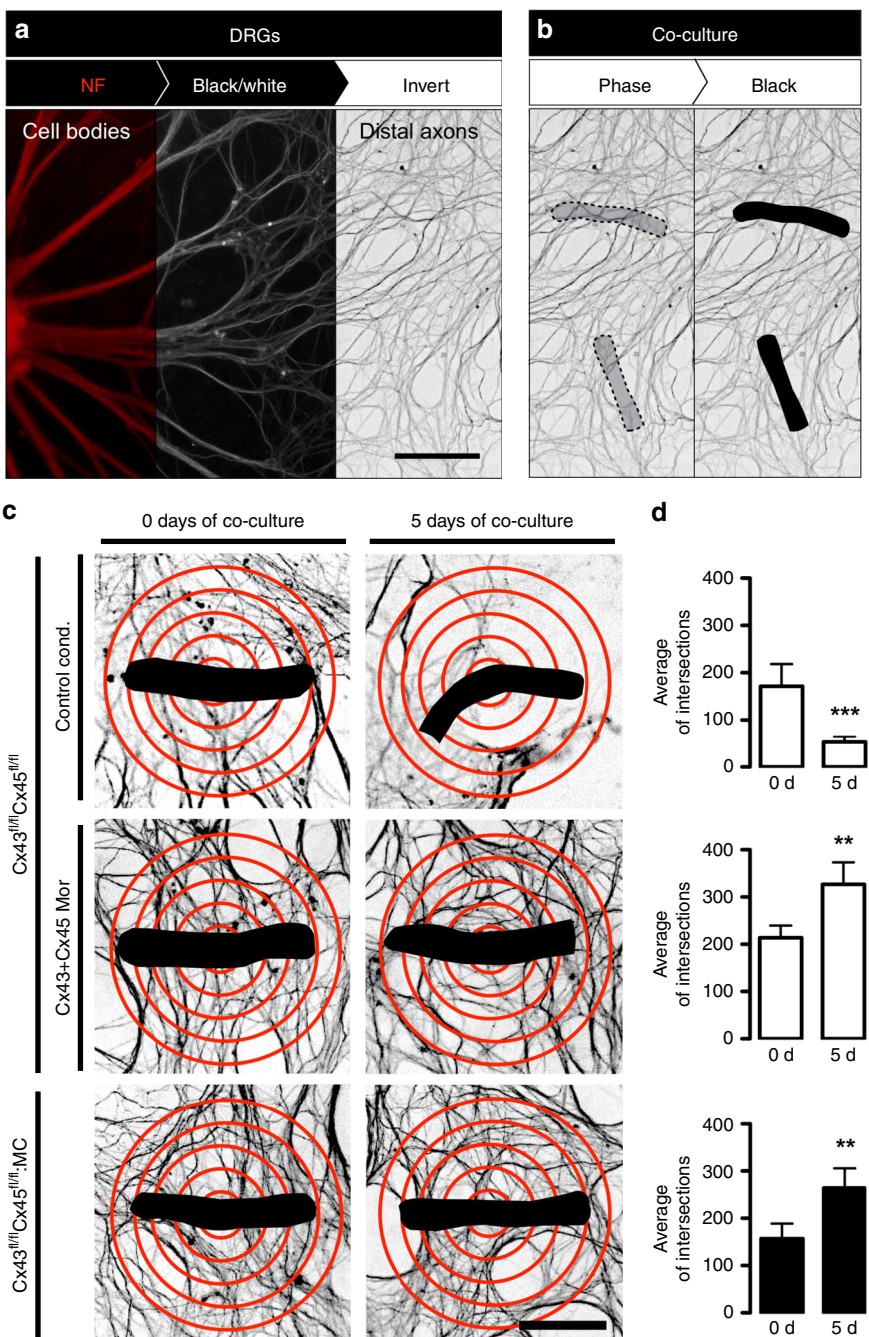

**Fig. 7 The absence of Cx43 and Cx45 in cultured skeletal myofibers increases the number of axons around the myofibers in co-culture. a** Immunofluorescence on DRGs using antibodies against neurofilament 200 (NF; red; left), converted to monochrome (black and white; middle), and inverted (white and black; right). Scale bar: 100 μm. **b** Localization of myofibers in co-culture using phase-contrast images (dotted figure; left), converted to black images (dotted figure; right). **c** Quantification of the number of intersections between distal axons and 5 red concentric circles spaced at 15 μm centered on the myofibers (black). DRGs co-cultured with myofibers from Cx43$^{fl/fl}$Cx45$^{fl/fl}$ mice in control conditions (Control cond.; top row), or in presence of Cx43 + Cx45 Mor (middle row), and Cx43$^{fl/fl}$Cx45$^{fl/fl}$:MC mice (bottom row) at 0 days (right column), or 5 days (left column) of co-culture. Scale bar: 50 μm. **d** Average of intersections. $N = 5$; at least 55 myofibers recorded in each independent experiment, each value is the mean ± SEM. The results obtained from Cx43$^{fl/fl}$Cx45$^{fl/fl}$ mice are represented by white bars and from Cx43$^{fl/fl}$Cx45$^{fl/fl}$:MC mice by black bars. ***$p < 0.001$, and **$p < 0.005$, for 5 days of co-culture compared with 0 days of co-culture, by Student's $t$ test.

muscles, which in future studies could be used to explore their possible benefits for reinnervation.

## Discussion
The findings of the present work indicate that binding to nAChR of ACh released from the terminal bouton provides the trophic signal that prevents de novo synthesis and incorporation of Cx HCs into the sarcolemma to initiate a sequence of deleterious changes that include permeabilization of the sarcolemma, increased cytosolic [Ca$^{2+}$] and [Na$^+$], and upregulation of atrogenes resulting in protein catabolism and muscle atrophy. These data provide insight regarding mechanisms by which a nerve stump delays the onset of electrophysiologic

perturbations of denervation in proportion to the length of the stump.

While the above data implicate signaling downstream of nAChR in suppression sarcolemmal Cx HCs expression, the molecular underpinnings responsible remain uncertain. Findings that denervation did not alter mRNA levels for Cx43 or Cx45 while significantly increasing protein levels for these Cxs suggest that regulation of Cx protein level within myofibers is under posttranscriptional control.

Further studies are needed to thoroughly understand the role (s) of intracellular signaling pathways downstream of nAChR in modulating the de novo incorporation of Cx HCs into the sarcolemma and the molecular mechanisms by which these signaling pathways regulate expression and trafficking of Cx HCs.

Findings that an uncontrolled opening of Cx HCs in the astrocytes of the brain causes the excessive release of molecules such as glutamate and ATP raise the possibility that accumulation of these molecules in the extracellular space could be toxic for neighboring cells[56]. To understand whether Cx HCs expression in the sarcolemma of denervated myofibers might influence reinnervation of denervated fibers, we also tested whether the activity of Cx HCs in the sarcolemma interferes with the neurite growth and muscular reinnervation using co-cultures of dorsal root ganglia (DRGs) explants and skeletal myofibers. The data support the conclusion that denervated muscle fibers suppress axon grew and reinnervation; it is now apparent that skeletal muscle releases a large number of cytokines, growth factors and metabolically active small metabolites[57–59]. One might speculate that one or more of these "myokines" is responsible for the deleterious effects of denervated myofibers on axon growth and innervation and represents attractive targets for therapies. Moreover, it seems likely that their release via a Cx HC-dependent mechanism is a crucial determinant of axon growth and reinnervation. It is noteworthy that small molecules including ATP[22,23], prostaglandins[60,61], and others small bioactive molecules are released from the cell through Cx HCs and exert autocrine or paracrine effects[24,25]. The co-culture system described herein provides a starting point for the identification of molecule(s) released by denervated myofibers and provide a feasible approach to identify the factors released by denervated myofibers, which in future studies could be used to develop strategies that favor reinnervation.

## Methods

**Reagents**. Mouse monoclonal anti-AchRɛ antibodies (1:100 dilution), and rabbit polyclonal anti-MyoD (1:100 dilution) were obtained from Santa Cruz Biotechnology, rabbit monoclonal anti-synaptophysin (1:1000 dilution) from Abcam (Cambridge, MA, USA), mouse monoclonal anti-Cx43 (1:1000 dilution) was obtained from Sigma-Aldrich, rabbit polyclonal anti-Cx45 (1:1000 dilution) and anti-neurofilament 200 (1:1000 dilution) was obtained from Thermo Fisher Scientific. Secondary antibody conjugated to Cy2 or Cy3 (1:200) were purchased from Jackson Immuno Research (Indianapolis, IN, USA). Morpholinos for mouse connexin43 (5′ TCTGGGCACCTCTCTTTCACRRAAT 3′) and mouse connexin45 (5′ TTGGTTTGCCCTGTTCACCAGAACT 3′) were obtained from Gene Tools, LLC (Philomath, OR, USA), Collagen I from rat tail, Leibovitz's L-15 Medium (L-15), Neurobasal medium, B-27 supplements, L-glutamine, monoclonal Anti-Neurofilament 200, Lipofectamine®2000, natural mouse NGF 7S (NGF 7S), recombinant human BDNF, ethidium bromide (Etd+), α-bungarotoxin-Alexa Fluor® 488, F-12 medium, DMEM/F-12 (Dulbecco's Modified Eagle Medium/ Nutrient Mixture F-12), fetal bovine serum (FBS), penicillin–treptomycin (Peni-Strepto), SBFI AM, cell permeant (SBFI), FURA-2, AM, cell permeant, 4′,6-dia-midino-2-phenylindole, dihydrochloride (DAPI), TRIzol® Reagent were obtained from Life Technologies (Grand Island, NY, USA). Dow Corning High-Vacuum Grease were purchased from Thermo Fisher Scientific. Fluoromount-G from Electron Microscopy Science (Hatfield, PA, USA,N-benzyl-p-toluene sulfonamide (BTS), collagenase type I rhodamine dextran (RD: 10 kDa), suramin sodium salt (suramin), lanthanum(III) chloride hepatahydrate (La³⁺), and cabenoxolone (Cbx) were obtained from Sigma (St. Louis, USA), pancuronium from InresaArzneimittel GmbH (Freiburg imBreisgau, Germany), Adenosine-5′-triphosphate (ATP), and adenosine-5′-0-(3-thiotriphosphate) (ATP-γ-S) were obtained from Roche

Diagnostics (Mannheim, Germany), GoTaq® Flexi DNA Polymerase kit from Promega (Madison, WI, USA), Krebs HEPES buffer (in mM: 145 NaCl, 5 KCl, 3 CaCl₂, 1 MgCl₂, 5.6 glucose, 10 HEPES-Na, pH 7.4). To qPCR Trim63-fw: GCTGAGTAACTGCATCTCCAT. Trim63-rv: GCTATTCTCCTTGGTCAC TCTG. Trim63-probe: FAM-AACGACCGAGTGCAGACGATCAT-BHQ1. Fbxo32-fw: TCTCCAGACTCTCTACACATCC. Fbxo32-rv: GAATGGTCTCCA TCCGATACAC. Fbxo32-probe: FAM-AGTCTGTGCTGGTGGGCAACATTA-B HQ1. Bcl2-fw: GTGGTGGAGGAACTCTTCAG. Bcl2-rv: GTTCAGGTACT CAGTCATCCAC. Bcl2-probe: FAM-TCATGTGTGTGGAGAGCGTCAACA-BH Q1. Bak1-fw: GAGAAGAGACCTGAGCACAATC. Bak1-rv: GTGAAGTAGGT-CATGCAGTCTC. Bak1-probe: FAM-CCCTACATTGGCTCCCAAGACCAA-BH Q1. Bnip3-fw: GACGAAGTAGCTCCAAGAGTTC. Bnip3-rv: CCAAAGCTGTG GCTGTCTAT. Bnip3-probe: FAM-CGCTCCCAGACACCACAAGATACC-BH Q1. Actb-fw: TTTCCAGCCTTCCTTCTTGG. Actb-rv: GGCATAGAGGTCTTT ACGGATG. Actb-probe: HEX-TGGAATCCTGTGGCATCCATGAAACT-BHQ2.

**Animals**. All procedures were approved by the Institutional Bioethics Committee of the Pontificia Universidad Católica de Chile (Protocol CBB-006/2013) and complied with National Institutes of Health guidelines. Male C57/Bl6 (C57), male Cx43<sup>fl/fl</sup>Cx45<sup>fl/fl</sup> (Cx43<sup>fl/fl</sup>Cx45<sup>fl</sup>), and male Cx43<sup>fl/fl</sup>Cx45<sup>fl/fl</sup>:MyoD-Cre (Cx43<sup>fl</sup>Cx45<sup>fl</sup>:MC) mice were used. P2X₇ mice were generously donated by Dr. Claudio Acuña from Universidad de Santiago de Chile. 57 mice of the same age were also used for comparison with (Cx43<sup>fl/fl</sup>Cx45<sup>fl/fl</sup>) mice, and since these were found to present no significant differences in all parameters studied in this work, we used (Cx43<sup>fl/fl</sup>Cx45<sup>fl/fl</sup>) mice as control animals in order to reduce the total number of euthanized animals. Skeletal myofibers were obtained from male mice of 2 month old. DRGs explants were obtained from day 14.5 mice embryos. All mice were maintained under light:dark 12:12 cycles with food and water ad libitum.

**Reverse transcription polymerase chain reaction (PCR)**. Total RNA was isolated from myofibers cultured using TRIzol following the manufacturer's instructions (Ambion). Two microgram aliquots of total RNA were transcribed to cDNA using MMLV-reverse transcriptase (Fermentas), and mRNA levels were evaluated by PCR amplification (GoTaq Flexi DNA polymerase; Promega). The PCR reaction was performed in 25 μL, containing 5 μL of 5× PCR buffer, 1.5 μL of Mg²⁺ 25 mM, 0.4 μL of 10 mM dNTPs, 0.4 μL of oligonucleotides. The oligos used were the following, Cx43 (34 cycles): S 5′-ATCCTTACCACGCCACCA-3′, AS 5′-CATTTT GGCTGTCGTCAGG -3′; Cx45 (34 cycles): S 5′-AAAGAGCAGAGCCAACCA-3′, AS-CCAAACCCTAAGTGAAGC-3′; GAPDH (26 cycles): S 5′-ACCACAGTCCA TGCCATCAC-3′, AS 5′-TCCACCACCCTGTTGCTGTA-3′; Panx7 (38 cycles): S 5′-CCACATCCGTCACAAGATAG-3′, AS 5′-TTCCTCGTCGTCCTCTTT-3′. The expected products were: Cx43 297pb; Cx45 311 bp; GAPDH 452; Pax7 352 bp.

**Isolation of skeletal myofibers**. Mouse skeletal myofibers from FDB were isolated using mechanical and enzymatic procedures. Briefly, muscles were carefully dissected and immersed in cultured medium (DMEM/F12 supplemented with 10% FBS and 1% penicillin/streptomycin) containing 2% collagenase type I, suramin 200 μM, and BTS 50 μM, then incubated for 1.5 h at 37 °C. Muscles were gently triturated using a Pasteur pipette to disperse single myofibers. Dissociated myofibers were resuspended in cultured medium with BTS.

**Culture of skeletal myofibers and drug treatment**. After isolation, skeletal myofibers were incubated in culture medium at 37 °C in 5% CO₂ for 24, 48, or 72 h. Different compounds were applied every 24 h. The compounds used were: carbachol (Cbc; 200 nM), methyl-carbamylcholine (Cmc; 200 nM), adenosine-5′-tri-phosphate (ATP, 500 μM), adenosine-5′-0-(3-thiotriphosphate) (ATP-γ-S; 500 μM), NGF (50 ng/mL), BDNF (50 ng/mL), carbenoxolone (Cbx; 100 μM), mor-pholinos against Cx43 plus Cx45 (Morph 43 + 45; 500 nM each) transfected with Lipofectamine using a 1:5 ratio, or pancuronium (Pcu; 150 nM). Recordings were made at 0, 24, or 48 h of culture.

**Dye uptake**. Functional state of connexin hemichannels in myofibers isolated from FDB was evaluated by Etd+ uptake[62,63]. Skeletal myofibers isolated were placed in Krebs HEPES buffer (mM: 145 NaCl, 5 KCl, 3 CaCl₂, 1 MgCl₂, 5.6 glucose, 10 HEPES-Na, 10 Tris pH 7.4) at room temperature and exposed to 5 μM Etd+ after which time-lapse fluorescence and snapshot images were taken. The Etd+ fluorescence was recorded in regions of interest that corresponded to nuclei of myofibers using a Nikon Eclipse Ti microscope (Japan).

**Intracellular free-Ca²⁺ and Na⁺ signal**. Cytoplasmic Ca²⁺ and Na⁺ signals were evaluated in FDB myofibers using the ratiometric dyes FURA-2 and SBFI, respectively. Myofibers were placed in DMEM/F12 without serum for 45 min at 37 °C and then, washed three times in Krebs-HEPES buffer. The experimental protocol for different imaging involved data acquisition at 340- and 380-nm excitation wavelengths. The ratio was obtained after dividing the 340-nm by the 380-nm fluorescence image on a pixel-by-pixel base ($\Delta = F340 \, nm/F380 \, nm$).

**RMP and MEPPs**. RMP measurements were recorded in vivo and in vitro, and MEPPs measurements were recorded in vivo in myofibers from FDB under whole-cell current clamp conditions at 25 °C. The pipette was a borosilicate electrode filled with a solution of 3 M KCl and the bath solution was Krebs-HEPES buffer, pH 7.4. The pipette resistance was ~50 MΩ. All experiments were performed using an Olympus IX 51 inverted microscope, with Axopatch1-D amplifier, Digidata1322 digitizer and Clampex 9.1 acquisition software. Data were analyzed with Clampfit 2.1 software.

**Unilateral hind limb denervation**. Under anesthesia with a mix 100 mg/kg ketamine and 10 mg/kg xylazine, a complete transection of the sciatic nerve in the left hind limb was done. The unoperated right hind limb was used as control. At post-operative day (PD) 1, 3, 5, 7, and 14 days, the TA muscles were carefully dissected from anaesthetized animals and different assays were performed. Then, animals were euthanized by cervical dislocation.

**Tenotomy**. Under anesthesia with a mix 100 mg/kg ketamine and 10 mg/kg xylazine, a complete transection of the Achilles tendon in the left hind limb was done. The unoperated right hind limb was used as control. After 7 days, the tibialis anterior (TA) muscles were carefully dissected from anaesthetized animals and cross-sectional area (CSA) measurements and Western blot were performed. Then, animals were euthanized by cervical dislocation.

**Western blot**. Gastrocnemius muscles were used instead of FDB or TO muscles because they yield enough protein for the western blot. Also, in mouse, gastrocnemius muscles have a similar percentage of fast myofibers to FDB muscles. Muscles were washed with ice-cold lysis buffer (in mM: 1 EDTA, 100 NaCl, 20 HEPES, and 1% Triton X- 100, pH 7.4) containing protease inhibitors (200 μg soybean trypsin protease inhibitor, 2 mM PMSF, 1 mg/mL benzamidine, 500 μg/mL leupeptin, and 1 mg/mL e-aminocaproic acid) and phosphatase inhibitors (in mM: 100 NaF, 20 $Na_4P_2O_7$, and 200 orthovanadate) and then frozen in liquid nitrogen. Muscles were minced in small pieces by using a razor blade and then homogenized (tissue homogenizer; Brinkmann) and sonicated (Heat Systems Microson). Tissue homogenates were centrifuged for 15 min at $13,000 \times g$, and pellets were discarded. Then, samples were processed for Western blot analyses of proteins. Blots were incubated overnight with appropriate dilutions of primary antibodies diluted in 5% fat-free milk–PBS solution. Then, blots were rinsed with 1% Tween 20 in PBS (TPBS) and incubated for 40 min at room temperature with HRP-conjugated goat anti-rabbit or anti-mouse IgGs (1:5000 dilution) (Santa Cruz Biotechnology). After five rinses with TPBS, immunoreactive proteins were detected with ECL reagents according to the manufacturer's instructions[15].

**Immunofluorescence analysis**. Myofibers isolated from FDB muscles were fixed with 4% formaldehyde for 10 min at room temperature. Next, muscles were incubated for 3 h at room temperature in blocking solution (50 mM $NH_4Cl$, 0.025% Triton, 1% BSA in PBS), then incubated overnight with appropriate dilutions of primary antibody, washed five times with PBS solution followed by 1 h incubation with secondary antibody, and mounted in Fluoromount G with DAPI. Immunoreactive binding sites were localized under a Nikon Eclipse Ti microscope.

**Myofiber CSA measurements**. The CSA of myofibers was determined in cross-sections of TA muscles fixed with 4% paraformaldehyde and stained with hematoxylin:eosin using ImageJ 1.46r software (National Institutes of Health).

**Molecular docking**. Docking were performed using AutoDock 4.0 software. The 3D structures of the compounds were sketched and optimized with the PM6 semi-empirical method using MOPAC2016, partial charges were assigned and rotatable bonds were identified. The receptor coordinates were extracted from cryo-electron microscopy structure of the *Torpedo* AChR (PDB ID: 2BG9)[64]. All files for docking calculations were prepared using AutoDock Tools (ADT). AutoDock uses a rapid grid-based method for energy evaluation. A grid volume enough to cover the entire surface of receptor was built ($240 \times 240 \times 320$ Å$^3$), using a grid spacing of 0.5 Å. The grid parameters were generated by AutoGrid 4.0 and the Lamarckian Genetic Algorithm (LGA) was used to perform a search of the conformational space of the ligands. Ten conformations for each compound were generated. The docking poses for all compounds were analyzed by examining their binding energy score. The most energetically favorable conformations were selected as the best poses.

**Compartmentalized DRGs explant cultures**. DRGs explants were obtained from day 14.5 mice embryos. Briefly, DRGs were dissected and placed in the central compartment (C) of Campenot chambers on Petri dishes coated with rat tail collage[65]. DRGs were maintained in Neurobasal medium supplemented with 2% B-27, 2 mM L-glutamine, 50 ng/mL NGF, 5% FBS and 1% Peni-Strepto (co-culture medium) at 37 °C in 5% $CO_2$, for 5 days, the co-culture medium was changed every 48 hours. DRG axons were allowed to grow under high-vacuum grease barriers to the lateral compartments (L), guided by superficial scratches.

**Co-culture of DRG explants and skeletal myofibers**. Freshly isolated FDB myofibers were placed in the lateral (L) compartments of Campenot chambers, over DRG distal axons at 5 days of culture. The co-culture was maintained in co-culture medium at 37 °C in 5% $CO_2$ for 5 days; the co-culture medium was changed every 48 h.

**Quantification of innervated myofibers**. At 5 days of co-culture, the cells were fixed and immunofluorescence microscopy using anti-neurofilament 200 and anti-AchRε antibodies was performed as described above. All myofibers were identified with phase-contrast images. On myofibers the colocalization between anti-neurofilament 200 and anti-AchRε was considered as innervated myofiber, the no colocalization was considered as no-innervated myofiber. The quantification of the innervated myofibers was performed by determining the percentage of innervated myofibers.

**Quantification of distal axons around the myofibers**. At 5 days of co-culture, the cells were fixed and immunofluorescence microscopy using anti-neurofilament 200 was performed, as described above. To facilitate the quantification of distal axons, the images were converted to monochrome and inverted. The myofibers were identified with phase-contrast and turned into black figures. To quantify the number of distal axons around the myofibers, 5 concentric circles centered on the myofibers spaced at 15 μm were placed, and the number of intersections between the distal axons and the circles was counted and was represented as the average of intersections. $N = 5$; at least 55 myofibers recorded in each independent experiment, each value is the mean ± SEM.

**Dye coupling**. The intercellular communication via gap junctions between HeLa-Cx43 cells (ATCC) was evaluated in subconfluent cultures (85%) by iontophoretic injection into one cell of 5% wt/vol Lucifer yellow (150 mM) in HCO$^-_3$ free F-12 medium buffered with 10 mM HEPES, pH 7.2 through glass microelectrodes. Briefly, coverslips containing cells were placed in a perfusion chamber and visualized in an inverted microscope (TE 200; Nikon, Melville, NY) equipped with xenon arc lamp and filters for Lucifer yellow (excitation wavelength 450–490 nm; emission wavelength above 520 nm). At 1 min after dye injection, surrounding cells were examined to determine whether dye transfer occurred. The incidence of dye coupling was calculated as the percentage of cases in which the dye transferred to at least one adjacent cell and the number of cells to which dye spread was determined and expressed as the index of dye coupling. In all experiments, the incidence of dye coupling was evaluated by injecting a minimum of ten cells.

**Data analysis and statistics**. Results were expressed as mean ± standard error of the mean (SEM). $N$ refers to the number of independent experiments. Pairs of means were compared using a Student's $t$ test. For multiple comparisons, one-way ANOVA was performed followed where significance was observed by Bonferroni post hoc test. Image analysis were carried out using Image-J (NIH) and the statistics analysis using Prism 8.0 software, GraphPad Inc. $p < 0.05$ was considered statistically significant.

**Reporting summary**. Further information on research design is available in the Nature Research Reporting Summary linked to this article.

## Data availability

The datasets generated during and/or analyzed during the current study are available from the corresponding author on reasonable request.

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

## Acknowledgements

J.C.S. acknowledges support from Fondo Nacional de Desarrollo Científico y Tecnológico (FONDECYT) grants 1111033, 1191329, and ICM-Economía grant P09-022-F from ICM-ECONOMIA, Chile. B.A.C. thanks FONDECYT grant 3170938 and CINV. C.V. acknowledges support from FONDECYT grant 11614338, BASAL Grant FB0807, and H2020-MSCA-RISE-2016 grant 734801 MAGNAMED. C.C. acknowledges support from the Department of Veterans Affairs, Rehabilitation Research and Development Service grant B-2020-C. We are also grateful to Teresa Vergara for their excellent technical assistance, to Egon Alvarez Vargas for assisting with molecular assays, and Dr. Claudio Acuña from Universidad de Santiago de Chile for providing the P2X$_7$ knockout mice.

## Author contributions

B.A.C. and J.C.S. designed research; B.A.C., A.A.V., C.P., P.F., C.F.L., R.E., M.F.M., C.V., L.A.C., E.B., H.G., and D.F.E. performed research; B.A.C., C.P.C., and J.C.S. analyzed data; B.A.C., C.P.C., and J.C.S. wrote the paper.

## Competing interests

The authors declare no competing interests.
