## [Peer Review File · Nature Communications]

Reviewers' Comments:

Reviewer #1:

Remarks to the Author:

The study by Cisterna and colleagues clearly demonstrate that connexin43/45 formed hemichannels are involved in denervation-induced muscle atrophy and that acetylcholine analogues can prevent both events and favour muscle reinnervation. This is quite an extensive study with a lot of interesting and clinically relevant findings. I have little to criticize but have a few suggestions/questions:

1. Is there a difference between fast and slow skeletal myofibers; which one were predominantly used in the present experiments?
2. Why did the authors use stable acetylcholine analogues to stimulate the myofibers and how was the concentration chosen (how does it relate to what is measured in vivo)? In vivo one has to assume variable interstitial acetylcholine concentrations rather than stable ones.
3. Both chronic and acute knockdown of both connexin 43 and 45 abolished the denervation-induced muscle atrophy. Are both connexins equally distributed throughout the outer membrane of myofibers or does one is preferentially located at gap junctions and becomes freely accessible only in cultured myofibers?

Reviewer #2:

Remarks to the Author:

This study aims to address a fundamental question in muscle biology: how does muscle denervation cause atrophy? The authors established an in vitro system to study the effects of active Acetylcholine receptor (AChR) on sarcolemma's permeability, membrane potential and fiber atrophy in isolated denervated mouse muscles. For this purpose they used different drugs to activate the AChR or to inhibit certain downstream kinase effectors (i.e. PKC). Unfortunately, the data do not yet permit definitive conclusions and several of the authors' conclusions require further biochemical study in vitro and especially in vivo. In addition, this study seems to follow several scientific directions (e.g. role of active AChR, role of Cx43 and Cx45, role of PKC, mechanisms of reinnervation of denervated muscle) but does not provide a definitive answer for most of them. Finally, the text contains no sound hypothesis, it lacks important details about the many molecules used (which should help the reader understand why certain experiments were done), the introduction is too short, and there is almost no discussion of the presented data.

Major points.

1. The in vitro model system has not been sufficiently validated for the investigation of denervation atrophy and thus conclusions about any observation should be made in a cautious manner. In order to validate that the isolated denervated muscle fibers studied in vitro recapitulate the genetic and biochemical alterations in denervated muscle in vivo the authors should determine the expression of several atrogenes (MuRF1, Atrogin1, autophagy genes, Bnip3; Gomes et al 2001, Bodine et al 2001, Sandri et al 2004), an increase in desmin phosphorylation (Volodin et al 2017), and a reduction in cross sectional area of muscle fibers (Many papers by Marco Sandri or Alfred L Goldberg). The data presented as Fig 1d is insufficient. In parallel, the authors should rule out the possibility that the denervated fibers grown in culture go through apoptosis rather than atrophy because several studies have shown that AChR can also regulate apoptosis (see review by Resende and Adhikari, 2009).
2. "In conclusion, the lack of nervous supply in cultured myofibers induces alterations equivalent to those observed in denervated muscles in vivo", this statement feels premature and an overstatement of the presented data.
3. The authors should explain clearly why ATP and ATP- γ -S were used in this study, as well as BDNF and NGF if there are no neurons in this in vitro system.
4. Fig 1c, 2d- why are the cbc and control fibers blue?

5. Fig 1d- a cbc control for 72hr is missing. Such control is important because the significant increase in atrogen1 and calpain can be observed only at 72hr (Fig 1d).
6. Fig 2: why treating isolated muscle fibers with an AChR blocker pancuronium if the muscles are denervated and the isolated fibers grown in culture lacking cholinergic neurons? It would probably be more valuable and informative to treat those fibers with Ach from 48hr in culture onward, when the effects on membrane potential and permeability are observed.
7. The conclusions overstate the presented data and more work should be done, especially in vivo. First, this paper lacks important controls such as an innervated muscle and denervated muscles from non-injected mice, which can be done only in vivo. For example, the pyri data in Figs 2f-g are not convincing - there seem to be a trend of atrophy in muscles from treated mice, and I am surprised the difference is not significant statistically given the shown SEM. In addition, a control of denervated muscle from untreated mice is missing. Finally, the logic behind planning such an experiment is unclear because sectioning of the sciatic nerve denervates most of the lower limb muscles and no Ach is secreted. What is the benefit then of treating mice with acetylcholinesterase inhibitor if there are no active neurons that secrete the neurotransmitter?
8. Fig 2i-j: it is unclear why the authors chose to inhibit those kinases and not others. Also, "with these results, we suggest that activation of protein kinase C is downstream of nAChR" - there is a question of novelty here because the connection between AChR and PKC has been well documented in the literature from 1998 (Nishizaki and Sumikawa). However, I was disappointed to learn that the authors did not follow up here on this interesting discovery in order to address the fundamental question, which they present as the purpose for this whole study: how does muscle denervation and lack of AChR activity promote atrophy? It will be important to document the potential mechanistic connection between PKC activation and Cx45 and Cx45 elevation in denervated muscle.
9. It is interesting that loss of both Cx43 and Cx45 is required to prevent denervation-induced atrophy (Fig 3 and Extended Data Fig. 7). The authors should discuss this point.
10. "We also evaluated the RMP in cultured myofibers (in vitro) and in denervated FDB muscles (in vivo)", the in vivo data are not presented. All studies involve isolation of muscle fibers and their maintenance in vitro. The authors have Cx43/Cx45 KO mice that allow more thorough investigation of the presented effects at the in vivo setting.
11. Fig 4f: unclear how the number of neuron-muscle contacts has been quantified.

Minor points.

1. Abstract: "denervation of skeletal muscle induces...accelerated protein catabolism", the authors are obviously not familiar with the heavy literature on muscle atrophy from the Alfred Goldberg's group who pioneered this field and is the expert on proteolysis in atrophying muscles (by denervation or other stimuli) because none of his paper were cited herein.
2. Labeling knockout mice as "Cx43fl/fl Cx45fl/fl:MC mice" is wrong and very confusing. They should be labeled as Cx43^{-/-} Cx45^{-/-} mice.
3. "The reduction in RMP was prevented by Cbc for 48 h, whereas it did not occur in FDB myofibers from Cx43fl/flCx45fl/fl:MC mice (Fig. 3d)" that is not what Fig 3d is showing. This is only one example. In several places in this paper there is a mismatch between the text and figures.

Reviewer #1 (Remarks to the Author):

The study by Cisterna and colleagues clearly demonstrate that connexin43/45 formed hemichannels are involved in denervation-induced muscle atrophy and that acetylcholine analogues can prevent both events and favour muscle reinnervation. This is quite an extensive study with a lot of interesting and clinically relevant findings. I have little to criticize but have a few suggestions/questions:

1. Is there a difference between fast and slow skeletal myofibers; which one were predominantly used in the present experiments?

Thank you very much for asking this relevant point. In fact, we did not explicitly say it, and now it is indicated in the text since, in most experiments, we used mouse *flexor digitorum brevis* (FDB) muscles which is a fast muscle and therefore contains mostly fast myofibers as demonstrated by Tarpey¹ (~96%). Only in the tenotomy experiments we used the gastrocnemius, which is also a fast muscle. The outcome of denervation on slow myofibers remains to be studied.

2. Why did the authors use stable acetylcholine analogues to stimulate the myofibers and how was the concentration chosen (how does it relate to what is measured in vivo)? In vivo one has to assume variable interstitial acetylcholine concentrations rather than stable ones.

Stable acetylcholine (ACh) was used because ACh is quickly hydrolyzed by acetylcholinesterase (AChE) with a half-life of ~0.16 ms² which could be a limitation to see any effect. The concentration of ACh used (200 nM) has been reported to activate the nicotinic ACh receptors in several skeletal muscle preparations^{3,4}, and the myofibers were also treated with BTS to avoid contraction and favor attachment on the tissue culture dish or coverslip to perform most measurements. We now performed experiments with myofibers treated with ACh applied every 6 hours for 72 hours of culture (Supplementary figure 1a) and found similar results as those obtained with stable ACh analogs (**Figure 1a, b** for Carbachol, and **Supplementary figure 1b, d** for Carbamylcholine). Thank you very much for bringing up this point.

3. Both chronic and acute knockdown of both connexin 43 and 45 abolished the denervation-induced muscle atrophy. Are both connexins equally distributed throughout the outer membrane of myofibers or does one is preferentially located at gap junctions and becomes freely accessible only in cultured myofibers?

Normal myofibers *in vivo* do not express connexins 43 or 45 or any other connexin. *In vivo* they are detectable after several days of denervation due to *de novo* expression. On the other hand, it has been previously shown that denervated skeletal muscles do not form gap junctions⁵. Therefore, it is not that they become available because of culture condition. With regard to their location of connexins, the functional assay of connexin hemichannels (Dye uptake, **Figure 5a, b**) indicates that the hemichannels are located in the sarcolemma. This assumption is consistent with our immunofluorescence detection of each connexin in culture myofiber shown in **figure 5c** where labeling appears striated.

Reviewer #2 (Remarks to the Author):

This study aims to address a fundamental question in muscle biology: how does muscle denervation cause atrophy? The authors established an *in vitro* system to study the effects of active Acetylcholine receptor (AChR) on sarcolemma's permeability, membrane potential and fiber atrophy in isolated denervated mouse muscles. For this purpose they used different drugs to activate the AChR or to inhibit certain downstream kinase effectors (i.e. PKC). Unfortunately, the data do not yet permit definitive conclusions and several of the authors' conclusions require further biochemical study *in vitro* and especially *in vivo*. In addition, this study seems to follow several scientific directions (e.g. role of active AChR, role of Cx43 and Cx45, role of PKC, mechanisms of reinnervation of denervated muscle) but does not provide a definitive answer for most of them. Finally, the text contains no sound hypothesis, it lacks important details about the many molecules used (which should help the reader understand why certain experiments were done), the introduction is too short, and there is almost no discussion of the presented data.

Major points.

1. The *in vitro* model system has not been sufficiently validated for the investigation of denervation atrophy and thus conclusions about any observation should be made in a cautious manner. In order to validate that the isolated denervated muscle fibers studied *in vitro* recapitulate the genetic and biochemical alterations in denervated muscle *in vivo* the authors should determine the expression of several atrogenes (MuRF1, Atrogin1, autophagy genes, Bnip3; Gomes et al 2001, Bodine et al 2001, Sandri et al 2004), an increase in desmin phosphorylation (Volodin et al 2017), and a reduction in cross sectional area of muscle fibers (Many papers by Marco Sandri or Alfred L Goldberg). The data presented as Fig 1d is insufficient. In parallel, the authors should rule out the possibility that the denervated fibers grown in culture go through apoptosis rather than atrophy because several studies have shown that AChR can also regulate apoptosis (see review by Resende and Adhikari, 2009).

We agree with all these observations. Consequently, we validate the culture system as a model of muscle atrophy by measuring the amount of mRNAs of molecular markers of atrophy, autophagy, and apoptosis. We also quoted several pioneer reports on muscle atrophy. The new data is shown in the new **figure 1**, and the following paragraphs were inserted to describe these findings.

To answer this question, we first validate the culture system as a model of muscle atrophy by measuring the amount of mRNAs of molecular markers of atrophy, autophagy, and apoptosis. To accomplish this purpose we used FDB myofibers cultured for 72 h undergo molecular changes compatible with those of the atrophy response observed *in vivo*, we evaluated the expression of atrophy genes called "atrogenes" (*Fbox32* and *Trim63*) and autophagy genes (*Bnip3*) which are altered *in vivo* in denervated FDB muscles^{6,7}. In addition, we evaluated the expression of a pro-apoptosis gene (*Bak1*) and compared it to the anti-apoptotic *Bcl2* gen^{8,9}. All genes were normalized as fold expression to β -actin.

In freshly isolated myofibers (*in vitro*, 0 h) the amounts of *FBox32*, *Trim63* and *Bnip3* were low and comparable to those of *in vivo* innervated (Inne) FDB muscles. However, the amount of these mRNAs showed a drastic increase in myofibers cultured for 72 h as well in FDB muscles after 7 days of denervation (Den) (**Fig. 1a**). These findings are consistent with pioneering research on gene expression of denervated skeletal muscle⁶. Similarly, the amount of *Bnip3* mRNA (**Fig. 1a**) measured in freshly isolated myofibers was similar

to that of innervated FDB muscles but increased significantly at 72 h in culture reaching values similar to those found *in vivo* after 7 days of denervation of FDB muscles (**Fig. 1a**), again these *in vitro* findings are representative of previous *in vivo* studies of the effects of denervation⁷. In contrast, the amount of mRNAs of *FBox32*, *Trim63* and *Bnip3* in myofibers treated with Cbc applied every day up to 72 h were comparable to the amount of each mRNA detected in innervated FDB muscles or in freshly isolated myofibers (**Fig. 1a**). In addition, we observed increased atrogin-1 and μ -calpain immunoreactivity (**Fig. 1b**) after 48 h and 72 h of culture under control conditions. These changes were not observed in myofibers treated with Cbc (**Fig. 1b**).

With regard to the amount of pro-apoptotic (*Bak1*) and anti-apoptotic (*Bcl2*) markers, we observed a decrease the amount of *Bak1* mRNA and an increase in *Bcl2* mRNA, which gives a decreased *Bak1/Bcl2* ratio, in Inne muscles, whereas an increase mRNA amount of both, *Bak1* and *Bcl2*, with a greater increase in *Bak1*, yields an increased *Bak1/Bcl2* ratio, in Den muscles as reported previously for denervated skeletal muscles^{8,9}. In cultured myofibers, we found a decrease amount of mRNA of both, *Bak1* and *Bcl2*, with a greater decrease in *Bak1*, giving a decreased *Bak1/Bcl2* ratio at 0 h. At 72 h of culture, we found a decrease amount of mRNA of *Bak1* and an increase in *Bcl2*, giving a decreased *Bak1/Bcl2* ratio. Myofibers treated for 72 h with Cbc showed a decrease in *Bak1* and an increase in *Bcl2* mRNAs, which gives a decreased *Bak1/Bcl2* ratio. In myofibers treated for 72 h with Paclitaxel used as positive control for apoptosis¹⁰, the amount of *Bak1* mRNA increased and that of *Bcl2* mRNA decrease, resulting in an increased *Bak1/Bcl2* ratio (**Fig. 1c, d**). In addition, we found similar caspase 3 and annexin V immunoreactivity at 0 h and after 72 h under control conditions, as well as after 72 h treatment with Cbc, whereas myofibers treated for 72 h with Paclitaxel showed an increase in caspase 3 and annexin V immunoreactivity (**Fig. 1e**). Thus, the above findings revealed that cultured myofibers undergo atrophy without apoptosis similar to what occurs *in vivo* denervated muscles, validating the primary culture of primary myofibers as a model of muscle denervation.

2. “In conclusion, the lack of nervous supply in cultured myofibers induces alterations equivalent to those observed in denervated muscles in vivo”, this statement feels premature and an overstatement of the presented data.

We agree with this comment, but with the new data included in the manuscript, we feel more confident that the statement is now correct. Thank you very much for your valuable critics.

3. The authors should explain clearly why ATP and ATP- γ -S were used in this study, as well as BDNF and NGF if there are no neurons in this in vitro system.

The objective of the trial was to evaluate the effect of different neuron-derived molecules that on myofibers, including ACh, ATP, BDNF, and NGF. To do this, we designed an *in vitro* model of denervation in which only the synaptic bouton is retained in the synaptic cleft. Since the source of neuron-derived molecules is very limited, the outcome of denervation appeared earlier than *in vivo* denervated myofiber where a longer nerve stalk remains attached to the myofiber and therefore can provide a different factor to the myofibers for a longer period of time. ATP and ACh are co-release upon neurosecretion. ATP- γ -S is just more stable than ATP that is quickly hydrolyzed and provide an alternative that could be more successful than using ATP. BDNF and NGF are released by neurons that are absent in culture. This issue is now explained at the beginning of the text as follows: Innervation by motor neurons is key to maintaining the normal phenotype of skeletal myofibers; denervation induces atrophy¹⁻³. How the motor neuron exerts its protective function on the myofiber is unknown. The studies reported here were, in large part, stimulated by observations that denervation performed at the sciatic notch leaves an axonal stump of several centimeters and that the ability of the stump to transmit impulses is prolonged by about 45 min for each additional centimeter in the axonal

stump, suggesting that there is a direct relationship between the length of the axonal stump and the period after denervation over which the nerve stump is capable of transmission of impulses to the muscle⁴. Similarly, the axonal stump length also is inversely related to the onset of muscular disorders resulting from denervation such as fibrillation and hypersensitivity to acetylcholine⁵. In addition, it has been shown that the terminal bouton releases small amounts of ACh in the absence of an action potential to result in MMEPs⁴. We speculated therefore that the stump supported release from the terminal bouton of ACh and other neurotransmitters for at least a short period and that such release in the absence of motor neuron or sarcolemmal action potentials might be involved in protecting myofibers against atrophy by suppressing *de-novo* expression of Cx hemichannels. Cx hemichannels are absent from the sarcolemma of innervated muscle; their expression in denervated muscle reduces resting membrane potential, increases sarcolemmal permeability to cations and small molecules, and upregulates muscle atrophy genes (atrogenes) such as *Fbox32* and *Trim63*.

4. Fig 1c, 2d- why are the cbc and control fibers blue?

Sorry for the insufficient explanation. The experiment is a measurement of free intracellular Ca²⁺ signal using FURA-2, a ratiometric indicator dye; the blue color indicates low Ca²⁺ signal. The pseudocolor scale is on the right side of each image. Now, this issue is explained in the legend of **Figure 3**.

5. Fig 1d- a cbc control for 72hr is missing. Such control is important because the significant increase in atrogin1 and calpain can be observed only at 72hr (Fig 1d).

We agree, and now it included the 72 h result indicating that is equivalent to the immunofluorescence at all times studied.

6. Fig 2: why treating isolated muscle fibers with an AChR blocker pancuronium if the muscles are denervated and the isolated fibers grown in culture lacking cholinergic neurons? It would probably be more valuable and informative to treat those fibers with Ach from 48hr in culture onward, when the effects on membrane potential and permeability are observed.

It is true that myofibers are denervated but, the synaptic bouton is still inserted in the neuromuscular junction as shown in **Supplementary Figure 5**. Therefore, the molecular machinery for choline re-uptake and synthesis of ACh is still present although for a limited time. These imply that agents promote retention of synaptic vesicles such as Pcu should block the excitatory miniature end-plate potentials (MEPPs) in the neuromuscular junction and advance the outcome of denervation (**Fig. 4a-d**). In contrast, if the compound can increase the half-life of ACh such as pyridostigmine (Pyri) bromide, an acetylcholinesterase inhibitor, should delay the appearance of denervation-induced outcome (**Fig. 4e-g**). We have tested the possibility of reverting the RMP, and we found that ACh added after 48 h of culture, when the RMP and sarcolemmal permeability are reduced and increased, respectively, but we did not find a reversion of the phenomenon. In our experiments, we observed that ACh could prevent but does not revert the denervation-induced outcome (**Supplementary Figure 3**).

7. The conclusions overstate the presented data and more work should be done, especially in vivo. First, this paper lacks important controls such as an innervated muscle and denervated muscles from non-injected mice, which can be done only in vivo. For example, the pyri data in Figs 2f-g are not convincing

- there seem to be a trend of atrophy in muscles from treated mice, and I am surprised the difference is not significant statistically given the shown SEM. In addition, a control of denervated muscle from untreated mice is missing. Finally, the logic behind planning such an experiment is unclear because sectioning of the sciatic nerve denervates most of the lower limb muscles and no ACh is secreted. What is the benefit then of treating mice with acetylcholinesterase inhibitor if there are no active neurons that secrete the neurotransmitter?

Several controls were omitted in the original version of the manuscript to reduce the size of the figures and have enough room for the text. We now included the requested controls in each figure. Thank you very much for requesting all these controls.

Figures 4f, g were enlarged and analyzed again. The requested controls were included: non-injected mice and Cx43 and Cx45 KO.

The logic behind the experiment is to increase the half-life of the remaking ACh in the denervated muscles, which is rapidly degraded by the action of the AChE (0.16 ms)², adding Pyridostigmine (an AChE inhibitor). The whole idea is based on manipulating the amount of spontaneously released ACh, not the amount of ACh released promoted by action potentials.

It is well established that denervation performed in the sciatic notch, leaves an axonal stump of several centimeters. It has been established that the ability of the stump to transmit impulses is prolonged by about 45 min for each additional centimeter in the axonal stump, suggesting that there is a direct relationship between the length of the axonal stump and the transmission of impulses to the muscle¹¹. Similarly, the axonal stump length also influences the onset of muscular disorders such as fibrillation and hypersensitivity to acetylcholine¹².

8. Fig 2i-j: it is unclear why the authors chose to inhibit those kinases and not others. Also, “with these results, we suggest that activation of protein kinase C is downstream of nAChR” – there is a question of novelty here because the connection between AChR and PKC has been well documented in the literature from 1998 (Nishizaki and Sumikawa). However, I was disappointed to learn that the authors did not follow up here on this interesting discovery in order to address the fundamental question, which they present as the purpose for this whole study: how does muscle denervation and lack of AChR activity promote atrophy? It will be important to document the potential mechanistic connection between PKC activation and Cx43 and Cx45 elevation in denervated muscle.

We agree that it would be of interest to study as many kinases as possible but this is not the main interest of this communication, and we will follow it in a future report since it requires a very different set of techniques that currently we do not perform in the laboratory. Alternatively, it could be that PKC inhibition increases the rate of desensitization of the ACh receptor as demonstrated by **Nishizaki and Sumikawa** and now quoted in this section of the manuscript. Since it might just be the result of enhanced desensitization, the result might not be so novel and could be just consistent with all the other approaches described above on the role of activation of ACh receptors.

9. It is interesting that loss of both Cx43 and Cx45 is required to prevent denervation-induced atrophy (Fig 3 and Extended Data Fig. 7). The authors should discuss this point.

An important point, thank you very much. We now inserted the following text. Myofibers deficient in Cx43 or Cx45 showed partial protection suggesting the involvement of both protein subunits that could form homomeric or heteromeric hemichannels which are permeable to ions including Na^+ , K^+ , and Ca^{2+} as well as small molecules such as fluorescent permeability probes¹³.

10. “We also evaluated the RMP in cultured myofibers (in vitro) and in denervated FDB muscles (in vivo)”, the in vivo data are not presented. All studies involve isolation of muscle fibers and their maintenance in vitro. The authors have Cx43/Cx45 KO mice that allow more thorough investigation of the presented effects at the in vivo setting.

Due to limit space, we did not include in the original version of the manuscript. The *in vivo* data are now presented in **Supplementary Figures 2, 11 and 12.**

11. Fig 4f: unclear how the number of neuron-muscle contacts has been quantified.

In the original version of the manuscript was explain in Methods. Quantification of distal axons around the myofibers” and “Quantification of innervated myofibers”.

Minor points.

1. Abstract: “denervation of skeletal muscle induces...accelerated protein catabolism”, the authors are obviously not familiar with the heavy literature on muscle atrophy from the Alfred Goldberg’s group who pioneered this field and is the expert on proteolysis in atrophying muscles (by denervation or other stimuli) because none of his paper were cited herein.

In the new version of the manuscript, Goldberg is now cited when appropriate. Thank you very much for the suggestion.

2. Labeling knockout mice as “Cx43fl/fl Cx45fl/fl:MC mice” is wrong and very confusing. They should be labeled as Cx43-/- Cx45-/- mice.

Genetic experts in our previous manuscript have previously corrected the nomenclature used (for example see Cea et al. 2016. Cell Mol Life Sci ¹⁴), and in order to avoid confusion in the literature by using different nomenclature, we decided to maintain the nomenclature used in the original version of the manuscript.

3. “The reduction in RMP was prevented by Cbc for 48 h, whereas it did not occur in FDB myofibers from Cx43fl/flCx45fl/fl:MC mice (Fig. 3d)” that is not what Fig 3d is showing. This is only one example. In several places in this paper there is a mismatch between the text and figures.

Thank you very much for noticing these problems. Now it Fixed up.

References

- 1 Tarpey, M. D. *et al.* Characterization and utilization of the flexor digitorum brevis for assessing skeletal muscle function. *Skelet Muscle* **8**, 14, doi:10.1186/s13395-018-0160-3 (2018).
- 2 Van der Kloot, W. & Molgo, J. Quantal acetylcholine release at the vertebrate neuromuscular junction. *Physiol Rev* **74**, 899-991, doi:10.1152/physrev.1994.74.4.899 (1994).
- 3 Kitazawa, T. *et al.* Muscarinic receptor subtypes involved in carbachol-induced contraction of mouse uterine smooth muscle. *Naunyn Schmiedebergs Arch Pharmacol* **377**, 503-513, doi:10.1007/s00210-007-0223-1 (2008).
- 4 Edelman, J. L., Kajimura, M., Woldemussie, E. & Sachs, G. Differential effects of carbachol on calcium entry and release in CHO cells expressing the m3 muscarinic receptor. *Cell Calcium* **16**, 181-193 (1994).
- 5 Cea, L. A. *et al.* De novo expression of connexin hemichannels in denervated fast skeletal muscles leads to atrophy. *Proc Natl Acad Sci U S A* **110**, 16229-16234, doi:10.1073/pnas.1312331110 (2013).
- 6 Bodine, S. C. *et al.* Identification of ubiquitin ligases required for skeletal muscle atrophy. *Science* **294**, 1704-1708, doi:10.1126/science.1065874 (2001).
- 7 Mammucari, C. *et al.* FoxO3 controls autophagy in skeletal muscle in vivo. *Cell Metab* **6**, 458-471, doi:10.1016/j.cmet.2007.11.001 (2007).
- 8 Adhihetty, P. J., O'Leary, M. F., Chabi, B., Wicks, K. L. & Hood, D. A. Effect of denervation on mitochondrially mediated apoptosis in skeletal muscle. *J Appl Physiol (1985)* **102**, 1143-1151, doi:10.1152/jappphysiol.00768.2006 (2007).
- 9 Siu, P. M. & Alway, S. E. Mitochondria-associated apoptotic signalling in denervated rat skeletal muscle. *J Physiol* **565**, 309-323, doi:10.1113/jphysiol.2004.081083 (2005).
- 10 Zhou, H. B. & Zhu, J. R. Paclitaxel induces apoptosis in human gastric carcinoma cells. *World J Gastroenterol* **9**, 442-445 (2003).
- 11 Miledi, R. & Slater, C. R. On the degeneration of rat neuromuscular junctions after nerve section. *J Physiol* **207**, 507-528 (1970).
- 12 Luco, J. V. & Eyzaguirre, C. Fibrillation and hypersensitivity to ACh in denervated muscle: effect of length of degenerating nerve fibers. *J Neurophysiol* **18**, 65-73, doi:10.1152/jn.1955.18.1.65 (1955).
- 13 Martinez, A. D., Hayrapetyan, V., Moreno, A. P. & Beyers, E. C. Connexin43 and connexin45 form heteromeric gap junction channels in which individual components determine permeability and regulation. *Circ Res* **90**, 1100-1107 (2002).
- 14 Cea, L. A. *et al.* Fast skeletal myofibers of mdx mouse, model of Duchenne muscular dystrophy, express connexin hemichannels that lead to apoptosis. *Cell Mol Life Sci* **73**, 2583-2599, doi:10.1007/s00018-016-2132-2 (2016).

Reviewers' Comments:

Reviewer #1:

Remarks to the Author:

None

Reviewer #2:

Remarks to the Author:

The authors did more work and added new convincing data showing the feasibility of cultured muscle fibers as an in vitro atrophy model. Although the fibers clearly atrophy, I am not convinced this is a reliable in vitro model for denervation; for example the mRNA levels of Cx43 and Cx45 are affected differently in this in vitro model compared with innervated vs. denervated muscles (Supplemental fig 6). Also the protein levels of Cx43 and Cx45 rise on denervation, but no data are provided for cultured myofibers.

In addition, many important controls that were missing in the previous version are now included, in vivo data was added, and the purpose of treating isolated muscle fibers with an AChR blocker was clarified.

I appreciate the new data on Cx43 and Cx45 mRNA that do not rise on muscle denervation, which indicate that these proteins are probably regulated post-translationally. Based on this finding one might speculate that AChR/PKC promote degradation of Cx43 and Cx45 in normal muscle and on denervation these proteins are stabilized (Supplemental Fig 6i-l). In my view more mechanistic insights here will be valuable and important since the authors already analyzed the effects of several kinases blockers " In order to identify the signaling pathway associated with the nAChR activity involved in the protective effect of ACh on denervated myofibers ..". Such information becomes even more important given the new data on immobilized muscle (Supplemental Fig 7) where signaling through the AChR is not inhibited and Cx43 and Cx45 protein levels do not rise compared with control. The data on immobilized muscles further support the hypothesis that AChR controls Cx43 and Cx45 protein levels.

Regulation by miRNA, which is discussed in the text of the revised manuscript, seems irrelevant here because the mRNA levels of Cx43 and Cx45 are similar in innervated and denervated muscles.

One experiment that can be done is to determine how Cx43 and Cx45 protein levels are affected in the presence of proteasome/lysosome inhibitors. It will probably be easier to work with the cultured myofibers in the presence of cbc and ask what happens to these proteins when proteasome inhibitors are added. Lysosome inhibitors should also be tested since Cx43 and Cx45 are located on the plasma membrane and may be recycled via the endosomal-lysosomal machinery.

Minor points:

1. Supplemental figure 6f-h - unclear from the figure what genes are tested
2. It is important to include the composition of lysis buffer and protocol for muscle homogenization in the methods section. It is unclear if the blots presented in Supplemental Fig 6i and k show analyses of whole cell extracts of membrane fractions (where we should expect to see Cx43 and Cx45). Knowing this will help understand if there is an overall increase in levels of these proteins or whether they only change their cellular distribution.

Reviewer #2 (Remarks to the Author):

1.- The authors did more work and added new convincing data showing the feasibility of cultured muscle fibers as an *in vitro* atrophy model. Although the fibers clearly atrophy, I am not convinced this is a reliable *in vitro* model for denervation; for example the mRNA levels of Cx43 and Cx45 are affected differently in this *in vitro* model compared with innervated vs. denervated muscles (Supplemental fig 6).

The differences in the relative amounts of Cx43 and Cx45 mRNAs might be explained by methodological differences because one was done using cultured myofibers (*in vitro*) and the other with whole muscles (*in vivo*). In **Supplementary Figure 6g and h** the mRNAs were isolated from cultured myofibers previously subjected to a protocol designed to generate myofibers free of satellite cells¹, which normally express connexins², whereas in **Supplementary Figure 6e and f** the mRNAs were isolated from whole muscles, which preferentially comprises muscle fibers, but also include endothelial cells, satellite cells, immune cells, among others that also express connexins. Furthermore, the differences might be accentuated by the high sensitivity of the RT-PCR technique.

2.- Also the protein levels of Cx43 and Cx45 rise on denervation, but no data are provided for cultured myofibers.

Although the amount of connexins could be evaluated by Western blot analysis, myofiber cultures from 6 mice did not provide enough protein for Western blot analysis. Therefore, to analyze the effect of lysosome and proteasome blockers on connexin protein levels in cultured myofibers, we used quantitative immunoreactivity of connexins. In addition, we evaluated the functional state of hemichannels using the dye uptake assay, which provides relevant information on the location of connexins at the sarcolemma. We found immunofluorescence and quantification of the reactivity of each protein, which is a procedure that reduces the number of sacrificed animals and provides sufficient information to confirm that the expected changes in levels of connexin proteins occurred; this approach also reduces the number of animals required since determination of protein amounts by Western blots in primary cultures requires more animals to obtain sufficient protein.

In addition, many important controls that were missing in the previous version are now included, *in vivo* data was added, and the purpose of treating isolated muscle fibers with an AChR blocker was clarified.

I appreciate the new data on Cx43 and Cx45 mRNA that do not rise on muscle denervation, which indicate that these proteins are probably regulated post-translationally. Based on this finding one might speculate that AChR/PKC promote degradation of Cx43 and Cx45 in normal muscle and on denervation these proteins are stabilized (Supplemental Fig 6i-l). In my view more mechanistic insights here will be valuable and important since the authors already analyzed the effects of several kinases blockers “ In order to identify the signaling pathway associated with the nAChR activity involved in the protective effect of ACh on denervated myofibers ..”. Such information becomes even more important given the new data on immobilized muscle (Supplemental Fig 7) where signaling through the AChR is not inhibited and Cx43 and Cx45 protein levels do not rise compared with control. The data on immobilized muscles further support the hypothesis that AChR controls Cx43 and Cx45 protein levels.

3.- Regulation by miRNA, which is discussed in the text of the revised manuscript, seems irrelevant here because the mRNA levels of Cx43 and Cx45 are similar in innervated and denervated muscles.

Thank you very much for this comment. We decided to remove from the text the discussion on miRNA.

4.- One experiment that can be done is to determine how Cx43 and Cx45 protein levels are affected in the presence of proteasome/lysosome inhibitors. It will probably be easier to work with the cultured myofibers in the presence of cbc and ask what happens to these proteins when proteasome inhibitors are added. Lysosome inhibitors should also be tested since Cx43 and Cx45 are located on the plasma membrane and

may be recycled via the endosomal-lysosomal machinery.

We performed the suggested experiments in *flexor digitorum brevis* myofibers at 0 and 48 h of culture, and using carbachol (Cbc), and blockers of lysosome (Chloroquine diphosphate salt), proteasome (G5, ubiquitin isopeptidase inhibitor I). We also evaluated the effect of cycloheximide, a protein synthesis blocker. We evaluated the immunoreactivity of Cx43 and Cx45 and the membrane sarcolemmal permeability to ethidium bromide. The concentrations of the compounds used were the same as those described by Mario Mauthe (2018)³ to 50 µg/ml Chloroquine, by Alessandra Fontanini (2009)⁴ to 1 µg/ml G5 and by Hannah Kaneb (2014)⁵ to 100 µg/ml cycloheximide. The results were described in a new paragraph in the manuscript, which is also included below; data are shown in a new figure, supplementary figure 7:

Moreover, we observed that blockade of lysosome- or ubiquitin-proteasome-dependent protein degradation pathway (chloroquine and G5 ubiquitin isopeptidase inhibitor, respectively) does not affect the increase in either Cx43/Cx45 immunoreactivities or sarcolemma permeability in cultured skeletal myofibers. However, blockade of protein synthesis with 100 µg/ml cycloheximide for 48 h prevents both alterations (**Supplementary Figure 7**), suggesting that activation of nicotinic AChRs lead to inhibition of protein synthesis.

Supplementary Figure 7. Blockade of lysosomes or 26S proteasomes does not prevent the increase in sarcolemma permeability and Cx immunoreactivity in cultured skeletal myofibers. Myofibers of *flexor digitorum brevis* muscles from Cx43^{fl/fl}Cx45^{fl/fl} mice were used. Cx43 and Cx45 immunofluorescence (Red: Cx43 or Cx45 immunoreactivity and blue: nuclei staining with DAPI, **a** to **f**) and membrane permeability evaluated as Etd⁺ uptake (**h** to **m**) at the indicated time points and treatment with carbachol (Cbc), 50 µg/ml chloroquine diphosphate salt (CQ; **d** and **k**), 1 µg/ml G5, a ubiquitin isopeptidase inhibitor I (G5; **e** and **l**), or 100 µg/ml cycloheximide (CHX; **f** and **m**). The fluorescence intensity of Cx immunoreactivity was quantified as Mean Fluorescence Intensity per µm³. N=6; at least five myofibers from each independent experiment were quantified, each value corresponds to the mean ± SEM. *** p < 0.001 (**g**). In dye uptake experiments, the first 5 min were recorded under basal conditions and the following 5 min cells were treated with 200 µM La³⁺ to block Cx HCs. **n**, Etd⁺ uptake rate during basal conditions. N=6; at least five myofibers from each independent experiment were recorded, each value corresponds to the mean ± SEM. *** p < 0.001, for the effect of La³⁺ compared to basal conditions by Student's *t* test. Scale bar: 50 µm.

Minor points:

1. Supplemental figure 6f-h - unclear from the figure what genes are tested

These figures show relative mRNA levels for Cx43 and Cx45 normalized to GAPDH. The legend for these figures has been revised to clarify what is shown as follows:

2. It is important to include the composition of lysis buffer and protocol for muscle homogenization in the methods section. It is unclear if the blots presented in Supplemental Fig 6i and k show analyses of whole cell extracts of membrane fractions (where we should expect to see Cx43 and Cx45). Knowing this will help understand if there is an overall increase in levels of these proteins or whether they only change their cellular distribution.

The protein fraction was obtained from whole-cell extracts, which indicates that the increase observed in western blots is an overall increase in the amount of Cxs. We have now indicated this in the Supplementary figure 6 legend as follows: “(i-l) Western blot analysis of Cx43 and Cx45 in whole innervated and 7 days post-denervated muscle.”

We added the buffer composition of the lysis buffer and protocol for muscle homogenization in the Methods section and references:

Western blot. Gastrocnemius muscles were used instead of FDB or TO muscles because they yield enough protein for the western blot. Also, in mouse, gastrocnemius muscles have a similar percentage of fast myofibers to FDB muscles. Muscles were washed with ice-cold lysis buffer (in mM: 1 EDTA, 100 NaCl, 20 HEPES, and 1% Triton X- 100, pH 7.4) containing protease inhibitors (200 µg soybean trypsin protease inhibitor, 2 mM PMSF, 1 mg/mL benzamidine, 500 µg/mL leupeptin, and 1 mg/mL e-aminocaproic acid) and phosphatase inhibitors (in mM: 100 NaF, 20 Na₄P₂O₇, and 200 orthovanadate) and then frozen in liquid nitrogen. Muscles were minced in small pieces by using a razor blade and then homogenized (tissue homogenizer; Brinkmann) and sonicated (Heat Systems Microson). Tissue homogenates were centrifuged for 15 min at 13,000 ×g, and pellets were discarded. Then, samples were processed for Western blot analyses of proteins. Blots were incubated overnight with appropriate dilutions of primary antibodies diluted in 5% fat-free milk-PBS solution. Then, blots were rinsed with 1% Tween 20 in PBS (TPBS) and incubated for 40 min at room temperature with HRP-conjugated goat anti-rabbit or anti-mouse IgGs (Santa Cruz Biotechnology). After five rinses with TPBS, immunoreactive proteins were detected with ECL reagents according to the manufacturer's instructions⁶.

References

1. Bischoff, R. Proliferation of muscle satellite cells on intact myofibers in culture. *Dev Biol* **115**, 129-39 (1986).
2. Ishido, M. & Kasuga, N. Characteristics of the Localization of Connexin 43 in Satellite Cells during Skeletal Muscle Regeneration In Vivo. *Acta Histochem Cytochem* **48**, 53-60 (2015).
3. Mauthe, M. *et al.* Chloroquine inhibits autophagic flux by decreasing autophagosome-lysosome fusion. *Autophagy* **14**, 1435-1455 (2018).
4. Fontanini, A. *et al.* The Isopeptidase Inhibitor G5 Triggers a Caspase-independent Necrotic Death in Cells Resistant to Apoptosis: A COMPARATIVE STUDY WITH THE PROTEASOME INHIBITOR BORTEZOMIB. *J Biol Chem* **284**, 8369-81 (2009).
5. Kaneb, H.M. *et al.* Deleterious mutations in the essential mRNA metabolism factor, hGle1, in amyotrophic lateral sclerosis. *Hum Mol Genet* **24**, 1363-73 (2015).
6. Cea, L.A. *et al.* De novo expression of connexin hemichannels in denervated fast skeletal muscles leads to atrophy. *Proc Natl Acad Sci U S A* **110**, 16229-34 (2013).

Reviewers' Comments:

Reviewer #2:

Remarks to the Author:

This paper has been substantially improved

The potential role of Acetylcholine receptor in regulation of protein synthesis is very interesting and I hope the authors will resolve the underlying mechanisms in future studies